# AANG: Automating Auxiliary learNING

**Lucio M. Dery**[1][*] **Paul Michel**[2] **Mikhail Khodak**[1] **Graham Neubig** [1] **Ameet Talwalkar**[1,3]
[1] Carnegie Mellon University [2] ENS PSL University [3] Hewlett Packard Enterprise

## Abstract

Auxiliary objectives, supplementary learning signals that are introduced to help aid learning on data-starved or highly complex end-tasks, are commonplace in machine learning. Whilst much work has been done to formulate useful auxiliary objectives, their construction is still an art which proceeds by slow and tedious hand-design. Intuition for how and when these objectives improve end-task performance has also had limited theoretical backing. In this work, we present an approach for automatically generating a suite of auxiliary objectives. We achieve this by deconstructing existing objectives within a novel unified taxonomy, identifying connections between them, and generating new ones based on the uncovered structure. Next, we theoretically formalize widely-held intuitions about how auxiliary learning improves generalization on the end-task. This leads us to a principled and efficient algorithm for searching the space of generated objectives to find those most useful to a specified end-task. With natural language processing (NLP) as our domain of study, we demonstrate that our automated auxiliary learning pipeline leads to strong improvements over competitive baselines across continued training experiments on a pre-trained model on 5 NLP tasks [1].

## 1 Introduction

The auxiliary learning paradigm, where we augment a primary objective with extra learning signals to boost end-task performance, is a staple of many machine learning (ML) domains. In natural language processing (NLP), well known models like SpanBERT (Joshi et al., 2020) and RoBERTa (Liu et al., 2019b) are trained on masked language modelling

| Objective | Data ($\mathcal{D}$) | Transform ($\mathcal{T}$) | Representation ($\mathcal{R}$) | Output ($\mathcal{O}$) |
|---|---|---|---|---|
| BERT | Out-of-domain | BERT-Op | Bidirectional | Denoise Token |
| TAPT | Task data | BERT-Op | Bidirectional | Denoise Token |
| DAPT | In-domain | BERT-Op | Bidirectional | Denoise Token |
| ELMO | Out-of-domain | No-Op | Left-to-Right and **Right-to-Left** | Next Token |
| GPT | Out-of-domain | No-Op | Left-To-Right | Next Token |
| XLNet | Out-of-domain | No-Op | Random factorized | Next Token |
| Electra | Neural LM Data | Replace | Bidirectional | Real / Synthetic |
| ⋯ | ⋯ | ⋯ | ⋯ | ⋯ |

Figure 1: We present the decomposition of some auxiliary objectives in NLP within our framework.

(MLM) auxiliary objectives (Devlin et al., 2018) before fine-tuning on the end-task. And for speech processing and reinforcement learning (RL), Oord et al. (2018) introduced the popular contrastive predictive coding objective which achieved state of the art performance in many settings when multi-tasked with the end-task. Despite these successes and many more, research into devising such objectives has progressed in a very local, objective-by-objective manner (Raffel et al., 2019; Clark et al., 2020; Grill et al., 2020; Chen et al., 2020). Auxiliary objectives are constructed by hand-design and without much overarching structure, relying on the experience and intuition of a select group of researchers versed at making appropriate design choices. Unfortunately, this status-quo not only creates a technical barrier of entry for exploring auxiliary objectives in new domains but also, by virtue of its incremental nature, limits the rate at which new objectives are discovered and investigated.

To address the above challenges, this paper presents a framework for automatically generating and utilizing a large set of candidate auxiliary objectives. Our framework is seeded by the following key observation: leading auxiliary objectives across multiple domains can be viewed as making different design decisions within a 4 stage pipeline: **Input Data** ($\mathcal{D}$) $\rightarrow$ **Input Transformation** ($\mathcal{T}$) $\rightarrow$

---

[*]Correspondence to : ldery@andrew.cmu.edu
[1]Code available at : https://github.com/ldery/Automating-Auxiliary-Learning.

**Model Representation** $(\mathcal{R}) \rightarrow$ **Output** $(\mathcal{O})$. For instance, in RL, a common auxiliary objective is to predict the environment's forward dynamics (Agrawal et al., 2016; Hafner et al., 2019). To construct this objective, the current task state-action pair $(\mathcal{D})$ is corrupted $(\mathcal{T})$ and then passed through the model to produce a latent representation $(\mathcal{R})$ which is finally used to predict the next state $(\mathcal{O})$. Similarly, in NLP, the XLNet (Yang et al., 2019) objective—which performs language modelling on a randomly factorized permutation of the input—can be written within our taxonomy as $\{\mathcal{D} =$ Out-of-Domain, $\mathcal{T} =$ No-op, $\mathcal{R} =$ Random-Factorized, $\mathcal{O} =$ Next Token$\}$. These two examples (along with others listed in Figure 1) fall within a class we term *named objectives*: objectives that have been previously proposed in the auxiliary learning literature.

Decomposing named objectives within our taxonomy provides a unified view of the auxiliary learning landscape. From this vantage point, it becomes clear that there are many unexplored combinations of the various primitives used across named objectives. This presents a simple formula for automatically generating a large set of candidate objectives: take the cartesian product of the design decisions across given stages (Figure 2). Using this compositional process,

| Data $(\mathcal{D})$ | | Transform $(\mathcal{T})$ | | Representation $(\mathcal{R})$ | | Output $(\mathcal{O})$ |
|---|---|---|---|---|---|---|
| Out-of-domain | | No-Op | | Bidirectional | | Next Token |
| In-domain | | Replace | | Left-to-Right | | Real / Synth |
| Task data | $\times$ | Mask | $\times$ | Right-to-Left | $\times$ | Denoise Token |
| Neural LM Data | | Noising embeds | | Rand. factorized | | TF-IDF |
| $\cdots$ | | $\cdots$ | | $\cdots$ | | $\cdots$ |

$$\downarrow$$

**TAPT** = {Task data $\rightarrow$ BERT-Op $\rightarrow$ Bidirectional $\rightarrow$ Denoise Token}
**GPT** = {Out-of-domain $\rightarrow$ No-Op $\rightarrow$ Left-to-Right $\rightarrow$ Next Token}
**New-Obj$_1$** = {Task data $\rightarrow$ BERT-Op $\rightarrow$ Left-to-Right $\rightarrow$ Denoise Token}
**New-Obj$_2$** = {In-domain $\rightarrow$ No-Op $\rightarrow$ Random Factorized $\rightarrow$ TF-IDF}
$\cdots$

Figure 2: Our framework in the context of NLP. We decompose named objectives within our four staged taxonomy : $\{\mathcal{D}, \mathcal{T}, \mathcal{R}, \mathcal{O}\}$. By taking the cartesian product of choices across stages, we reproduce named objectives and discover new ones.

not only can we reconstruct existing named objectives, we can also generate new combinations. This overcomes the tedium of implementing each objective independently since we can just reuse a small set of simple stage-wise primitives.

Generating a large set of objectives raises the natural question of how to efficiently select the most helpful ones for a given end task. Instead of leaving this to practitioner intuition, we develop principled guidelines to address this question by theoretically studying the impact of auxiliary learning on a particular end-task. Specifically, using arguments based on algorithmic stability (Hardt et al., 2016; Bousquet & Elisseeff, 2002), we derive end-task generalization error bounds that are dependent on the choice of auxiliary task. This contributes to existing theory (Saunshi et al., 2020; Xie et al., 2021) on how auxiliary learning impacts the end-task by suggesting a new candidate mechanism: auxiliary learning results in more stable optimization end-points in the sense of Bousquet & Elisseeff (2002), which in theory improves generalization of the final model.

Guided by our theory, we introduce **AANG** (**A**utomating **A**uxiliary Learni**NG**), an efficient, structure-aware algorithm for adaptively combining a set of related objectives to improve generalization on a specific end-task. **AANG** incorporates the following prescriptions from our theory: (i) auxiliary tasks that are more similar to the end-task are desirable. Given a set of objectives, **AANG** learns adaptive weights to bring the composite objective closer to the end-task; (ii) in general, more auxiliary data is better. **AANG** maximizes the effective amount of data used in training by using all the generated objectives instead of taking task-specific subsets.

To empirically validate our method for automatically generating and utilizing auxiliary objectives, we experiment on five NLP tasks. We do so in the widely-used setting of *continued pre-training* (Gururangan et al., 2020; Aghajanyan et al., 2021; Dery et al., 2021b; Zhang et al., 2022), where a model trained with a single auxiliary objective on large-scale data is further trained on end-task related data. Without introducing any external data or architectural modifications, variants of **AANG** outperform strong and widely used baselines in 4 out of 5 tasks. **AANG** achieves an average improvement of $\mathbf{4.2}\%$ over standard fine-tuning of RoBERTa across our chosen tasks. We believe our results will spur further research into exploring automating auxiliary learning across a variety of settings. Notably, while we focus on NLP when discussing the space of auxiliary objectives (Section 3) and in our empirical evaluation (Section 6), our theoretical results (Section 4) and **AANG** itself are domain-agnostic[2].

---

[2]Our ideas could be applied to domains like RL or computer vision (CV), where a similar dissection of existing objectives can be performed.

## 2 RELATED WORK

To properly scope this work, we define *auxiliary learning* as training a model on alternative objectives with the goal of improving performance on some primary end-task. Auxiliary learning is an instantiation of transfer learning (Caruana, 1997; Baxter, 2000; Ruder et al., 2019). It covers the pretrain-then-finetune paradigm (Huh et al., 2016; Devlin et al., 2018; Schneider et al., 2019; Gururangan et al., 2020) as well as end-task aware multitasking approaches (Lin et al., 2019; Dery et al., 2021a;b). Whilst auxiliary objectives may be meta-learned (Liu et al., 2019a; Navon et al., 2020), for simplicity – since incorporating these would require further complication of our design space – such objectives are out of the scope of this paper.

This work bears many parallels to the area of neural architecture search (NAS) (Stanley & Miikkulainen, 2002; Zoph & Le, 2016; Roberts et al., 2021). Whilst we seek to automate auxiliary learning, the objective of NAS is to automate the discovery of the right neural architecture given a specific end-task. Search spaces of candidate architectures are created by taking the cartesian product of architecture design choices across the depth of the network. The design of suitable architectural search spaces for a variety of settings has been an active area of research (Tan & Le, 2019; Howard et al., 2019; Dao et al., 2020; Roberts et al., 2021). To develop **AANG**, we borrow ideas from the NAS literature on efficient algorithms for sifting through spaces of architectures. Mirroring the popular differentiable NAS method DARTS Liu et al. (2018), we perform a continuous relaxation over the search space of objectives, allowing for efficient search by gradient descent. We also use a factored approach to model relationships between objectives that share primitives. This is inspired by recent work on stochastic-relaxation weight sharing (Dong & Yang, 2019; Li et al., 2020).

As a theoretical contribution, this work derives an end-task aware generalization error bound for auxiliary learning. Our bound is built on that of Hardt et al. (2016), who derive generalization bounds for parametric models trained with stochastic gradient descent (SGD). To derive their bounds, they leverage the concept of algorithmic stability introduced by Bousquet & Elisseeff (2002). Informally, a randomized algorithm is *uniformly stable* if changing a single training data point in the given samples does not change its end-point *too much*. Said change is characterized as the average difference in predictions between the two learned models. Stability implies generalization in expectation (Hardt et al., 2016; Kuzborskij & Lampert, 2018).

## 3 AUTOMATICALLY GENERATING AUXILIARY OBJECTIVES

To begin, we take a high-level view of the landscape of named objectives. Using running examples from NLP, we propose the following coarse structure for the sequence of choices made in the hand-design of auxiliary objectives:

1. **Data**, $\mathcal{D}$: Auxiliary objective pipelines begin with a choice of input data. Here, options can range from heterogeneous *out-of-domain* data (Radford et al., 2019), *in-domain* data with respect to the final end-task (Beltagy et al., 2019) or the *task data* itself (Gururangan et al., 2020). It may even include data outside the modality of the end-task.
2. **Input-Transformation**, $\mathcal{T}$: Many auxiliary objectives are self-supervised with respect to their input data. They corrupt or transform the input and then reconstruct it in whole or part. For example, input text tokens can be *masked*, *replaced* or *deleted*. Operations can also be aggregated as in *BERT-Op*: mask 80% of selected tokens and randomly replace 50% of the remaining Devlin et al. (2018); Liu et al. (2019b).
3. **Representation**, $\mathcal{R}$: After transformation, representations of the input data can be computed from a given model in different ways. A chosen token's representation can depend on only its left context (*Left-to-Right*) (Radford et al., 2018) or its right context (*Right-to-Left*) (Peters et al., 2018). It could also depend on the representations of a randomly selected permutation of other tokens (*Random Factorized*) Yang et al. (2019).
4. **Output**, $\mathcal{O}$: Finally, representations obtained from the previous stage are fed into a loss function producing a final output. The choice of output loss is usually coupled with the choice of transformation made in stage 2. Choices include but are not restricted to *denoising tokens*, *predicting the next token* or *predicting the TF-IDF* (Term Frequency-Inverse Document Frequency) of a token.

The above taxonomy $\{\mathcal{D} \to \mathcal{T} \to \mathcal{R} \to \mathcal{O}\}$ is expansive enough to cover a range of named auxiliary objectives of interest in NLP (Figure 1)[3]. For example, we can write any member of the GPT series (Radford et al., 2018; 2019; Brown et al., 2020) which perform left-to-right language modelling on out-of-domain data as $\{\mathcal{D} = \text{Out-of-Domain}, \mathcal{T} = \text{No-op}, \mathcal{R} = \text{Left-To-Right}, \mathcal{O} = \text{Next Token}\}$. We can summarize the pre-existing choices within each design stage to obtain a unique set of options. For example, we can reduce the set of model representation types used by the objectives enumerated in Figure 1 to the unique set $\mathcal{R} = \{\text{Bi-directional, Left-To-Right, Right-To-Left, Random-Factorized}\}$. Having summarized the list of primitives within each stage, a simple formula for generating a space of auxiliary objectives becomes apparent: take the cartesian product of the design choices at each stage (see Figure 2). In general, given an instance of our taxonomy, we can construct a space of objectives $\mathcal{A} = \mathcal{D} \times \mathcal{T} \times \mathcal{R} \times \mathcal{O}$ of size $|\mathcal{A}| \leq |\mathcal{D}| \times |\mathcal{T}| \times |\mathcal{R}| \times |\mathcal{O}|$. Consider $\text{New\_Obj}_1$ from Figure 2. This previously unexplored objective can be obtained by combining the special masking operation from BERT (*BERT-Op*) with computing model representations based on left-to-right causal masking as in GPT. In fact, this objective proved one of the most useful ones in our experiments below (see Figure 5).

Our framework also allows us to reason about whole families of objectives, $\mathcal{F}$, by thinking in terms of design stages and choices. For example, given a particular end-task $\mathbf{E}$ with input text $\mathbf{E}_{\mathcal{D}}$, we can create a family of objectives based solely on task data by fixing to that option in our input data stage; we call this family $\mathcal{F}_{\mathcal{D}=\mathbf{E}_{\mathcal{D}}}$. $\mathcal{F}_{\mathcal{D}=\mathbf{E}_{\mathcal{D}}}$ not only includes pre-existing TAPT Gururangan et al. (2020) but also unexplored objectives like task-data dependent variants of XLNET, ELMO etc. Auxiliary learning with $\mathcal{F}_{\mathcal{D}=\mathbf{E}_{\mathcal{D}}}$ can be seen as a relaxed form of data augmentation which we dub **task augmentation**. Whilst data augmentation requires applying transformations that preserve the data-point's label, task augmentation has no such restriction and thus offers greater flexibility in terms of specifying $\{\mathcal{T}, \mathcal{R}, \mathcal{O}\}$. We can also reason about expanding particular stages to include new primitives. Any supervised loss can be added to the output stage, $\mathcal{O}$, allowing us to potentially explore auxiliary objectives based on supervised signals like NER or POS tagging (Carreras et al., 2003; Charniak, 1997). A special example is setting $\mathcal{O}$ to the end-task supervised output $\mathbf{E}_{\mathcal{O}}$. This leads to $\mathcal{F}_{\mathcal{D}=\mathbf{E}_{\mathcal{D}}}^{\mathcal{O}=\mathbf{E}_{\mathcal{O}}}$ which is a subset of $\mathcal{F}_{\mathcal{D}=\mathbf{E}_{\mathcal{D}}}$. $\mathcal{F}_{\mathcal{D}=\mathbf{E}_{\mathcal{D}}}^{\mathcal{O}=\mathbf{E}_{\mathcal{O}}}$ includes many objectives like predicting the end-task signal from corrupted input data. In Section 6, we will introduce a search space of objectives that leverages task augmentation.

## 4 The Impact of Auxiliary Learning on End-task Generalization

In this section, we relieve reliance on practitioner intuition by deriving a set of guiding principles on how to effectively utilize the automatically generated objectives from Section 3.

Auxiliary learning influences the end-task through both training and generalization error. Previous theory has largely focused on characterizing the impact on end-task training error. Liu et al. (2021), for example, show that end-task agnostic pre-training can create a performance gap in training error compared to training with the end-task alone. The size of this gap depends on how dissimilar the pre-training auxiliary objective is from the end-task. They introduce the following assumption (which we will borrow) to formalize their notion of task similarity:

**Assumption A.1:** Let $f_e$ represent the end-task objective and $f_a$ be the auxiliary objective. There exists $\Delta \geq 0$ such that $\|\nabla f_a(\theta) - \nabla f_e(\theta)\| \leq \Delta \ \forall \theta$.

Note that $\theta$ represents all the parameters of the model. Smaller $\Delta$ implies $f_a$ is more similar to the primary task $f_e$. Liu et al. (2021) bound the end-task agnostic training error gap to be logarithmic in $\Delta$.

Unlike training error, end-task generalization error has gone unstudied in the auxiliary learning setting. Bounding the generalization error not only adds to our theoretical understanding of the impact of auxiliary learning but also provides insights to guide algorithm design. To arrive at a bound, we adapt the technique of Hardt et al. (2016) who derive a generalization bound on training with **only the end-task** via stochastic gradient descent. We consider the end-task aware setting where the end-task is multi-tasked with the auxiliary objective. This setting has recently been shown to improve end-task performance over the pretrain-then-finetune paradigm (Dery et al., 2021a;b; Yao et al., 2021).

**Auxiliary learning with Dynamic Sampling:** We are given an auxiliary objective $f_a(\cdot; z) \in [0, 1]$ with $N_a$ samples $S_a = (z_1, \ldots, z_{N_a})$ from the distribution $\mathcal{D}_a$. $f_a$ can either be a single objective or

---

[3]Although this taxonomy is quite expansive, it obviously does not consider other elements of objective creation such as choice of model architecture, optimizer settings, etc.

a weighted linear combination of objectives : $f_a = \sum_k w^k f_a^k$. At any iteration of SGD, we sample a choice of the end-task function $f_e$ or the auxiliary objective $f_a$ according to the probabilities $\lambda_e$, $\lambda_a \in [0, 1] \mid \lambda_e + \lambda_a = 1$. Given the chosen objective, we sample a data-point and perform stochastic gradient descent based on the sampled data-point. We now present our bound in the setting described.

**Theorem 4.1** (Auxiliary learning with Dynamic Sampling). *Assume that $f_e(; z_e), f_a(; z_a) \in [0, 1]$ are both L-Lipschitz with $\beta_e$ and $\beta_a$-smooth loss functions respectively. Consider that we have $N' = N_e + N_a$ total samples where $f_e$ and $f_a$ have $N_e$ and $N_a$ samples respectively. $r_e = \frac{N_e}{N'}$ is the fraction of the available data represented by the end-task. Suppose that we run stochastic gradient descent for T steps with monotonically non-increasing step sizes $\alpha_t \le \frac{c}{t}$ by dynamically sampling the tasks according to $\lambda_e$ and $\lambda_a$. Then, with respect to $f_e$, the generalization error is bounded by:*

$$\epsilon_{\text{gen}} \lesssim (\Delta)^{\frac{1}{1+c\lambda^*\beta^*}} \left(\frac{\gamma T}{N'}\right)^{1 - \frac{1}{c\lambda^*\beta^*+1}} \quad Where \quad \gamma = \frac{\lambda_e}{r_e} \tag{1}$$

*Here $\beta^* = \min\{\beta_e, \beta_a\}$ and $\lambda^*$ is the weighting of the function with smaller smoothness.*

*Proof.* See Appendix E for full proof and Appendix F for more discussion □

As a detailed inspection of the proof will show, we derive Equation 1 by appealing to algorithmic stability (Bousquet & Elisseeff, 2002; Hardt et al., 2016; Kuzborskij & Lampert, 2018) (Section 2). To our knowledge, ours is the first work to present an algorithmic stability view to formally explain how auxiliary learning influences end-task performance. Equation 1 surfaces the following prescriptions about learning with auxiliary tasks :

P1 Smaller $\Delta$ improves $\epsilon_{\text{gen}}$. This implies that the more similar the auxiliary objective is to the end-task (under Assumption A.1), the lower the generalization error.

P2 Larger $N'$ leads to smaller $\epsilon_{\text{gen}}$[4]. Since we usually have a fixed amount of task data $N_e$, we can increase $N'$ by adding more auxiliary data $N_a$.

## 5 END-TASK AWARE SEARCH OF STRUCTURED OBJECTIVE SPACES

---
**Algorithm 1 AANG**

**Input:** Search Space - $\mathcal{A}$
Factor vectors - $\{W^{\text{All}}, W^{\mathcal{I}}, W^{\mathcal{T}}, W^{\mathcal{R}}, W^{\mathcal{O}}\}$
End-task - **E**, End-task weight - $\lambda_e$
Initial Model Params - $\theta_0 \in \mathbf{R}^D$
**repeat**
  Sample a batch of $n$ objectives
  $\mathcal{K}^n \sim \mathcal{A}$
  Weighting of objectives in $\mathcal{K}^n$
  Construct $\mathbf{w}^n$
  **for** $k = 1$ **to** $n$ **do**
    $(d, t, r, o) = [\mathcal{K}^n_k].\text{stages}$
    $w^k \propto \exp\left(W^{\text{All}}_{(d, t, r, o)} + W^{\mathcal{I}}_d + W^{\mathcal{T}}_t + W^{\mathcal{R}}_r + W^{\mathcal{O}}_o\right)$
    $\mathbf{w}^n_k \leftarrow w^k$
  **end for**
  Get losses from batches of data
  $\hat{\mathcal{L}}_{\mathcal{A}}(\mathcal{K}^n, \mathbf{w}^n) = \sum_{k=1}^n w^k \mathcal{L}_k$
  $\mathcal{L}_{\text{total}} = \lambda_e \mathcal{L}_E + (1 - \lambda_e)\hat{\mathcal{L}}_{\mathcal{A}}$
  Get gradients and update factors
  $\theta_{t+1}, \{\nabla_{\mathbf{w}^n, \lambda_e}\} \leftarrow \text{META-TARTAN}(\theta_t, E, \mathcal{L}_{\text{total}})$
  Update $\{W^{\text{All}}, W^{\mathcal{I}}, W^{\mathcal{T}}, W^{\mathcal{R}}, W^{\mathcal{O}}\}$ using $\nabla_{\mathbf{w}^n}$
  Update $\lambda_e$ using $\nabla_{\lambda_e}$
**until** done
**Return :** $\theta_T$

---

Guided by Section 4, we build a practical method for exploring a set of objectives, $\mathcal{A}$.

Whilst the dynamic sampling setting described in Section 4 is amenable to theoretical consideration, we make a few practical changes to it. First, instead of performing alternating gradient descent by sampling $f_a, f_e$ according to $\lambda_e, \lambda_a$, we instead use them as multitask weights and perform joint training. Joint training has been found to produce superior results compared to alternating optimization when leveraging auxiliary objectives (Aghajanyan et al., 2021). We perform gradient descent on the following total loss which interpolates between the end-task and the auxiliary loss $\mathcal{L}_{\text{total}} = \lambda_e \mathcal{L}_E + (1 - \lambda_e)\mathcal{L}_{\mathcal{K}}$. Here, $\mathcal{K}$ is a chosen subset of $\mathcal{A}$.

Second, as indicated in Section 4, given $\mathcal{K}$, we can write the set as a single objective $f_a = \sum_{k \in \mathcal{K}} w^k f_a^k$. By Prescription P1, we want to choose $\{w^k\}$ such that $f_a$ has a small $\Delta$ with the end-task $f_e$. We would also like to set $\lambda_e$ such that the bound on $\epsilon_{\text{gen}}$ is minimized. Whilst a closed form exists for the optimal weightings $\lambda_e, \{w^k\}$, it depends on variables like $\{\Delta^k\}, \{\beta_a^k\}, L$ that are hard to estimate.

---
[4]This holds at fixed $\gamma$ which we achieve by adjusting $\lambda_e$ to account for introducing more auxiliary data.

We therefore propose to learn $\lambda_e, \{w^k\}$ in an online, data-driven way. To do this, we build on top of the META-TARTAN algorithm proposed by Dery et al. (2021b). META-TARTAN is a meta-learning algorithm that learns adaptive weights for different auxiliary tasks in a way that prioritizes end-task generalization. It learns $\{w^k\}$ by minimizing the loss on the end-task validation set: $\frac{\partial \mathcal{L}_{\mathbf{E}}^{val}}{\partial w^k} \approx -\left(\nabla_\theta \mathcal{L}_{f_a^k}\right)^T \left(\nabla_\theta \mathcal{L}_{\mathbf{E}}^{val}\right)$. This corresponds to learning $\{w^k\}$ such that $\left(\nabla_\theta f_a\right)^T \left(\nabla_\theta f_e\right)$ is maximized. This minimizes one of the terms that contributes to $\Delta$ and thus attempts to fulfil Prescription P1. We can similarly learn $\lambda_e$ to minimize the end-task validation loss. For a more detailed discussion of META-TARTAN, please see Appendix B.

So far, we have introduced independent weights, $\{w^k\}$, for each objective. This is sufficient in the case of unrelated objectives. However, the objectives in $\mathcal{A}$ share an underlying structure. We recognize this by using a factored approach to model each $w^k$. We introduce a factor vector for each of the 4 stages introduced in Section 3: $W^{\mathcal{D}} \in \mathbf{R}^{|\mathcal{D}|}, W^{\mathcal{T}} \in \mathbf{R}^{|\mathcal{T}|}, W^{\mathcal{R}} \in \mathbf{R}^{|\mathcal{R}|}$ and $W^{\mathcal{O}} \in \mathbf{R}^{|\mathcal{O}|}$. This ties together the weights of objectives that share primitives in common. To capture the fact that an objective can be more than the sum of it parts, we also introduce an independent weight for each objective : $W^{\mathrm{All}} \in \mathbf{R}^{|\mathcal{D}| \times |\mathcal{T}| \times |\mathcal{R}| \times |\mathcal{O}|}$. Consider the objective $k$ which is generated by the composition of the operations $\{d \in \mathcal{D}, \ t \in \mathcal{T}, \ r \in \mathcal{R}, \ o \in \mathcal{O}\}$, its weighting is computed as : $w^k \propto \exp\left(W_{(d,t,r,o)}^{\mathrm{All}} + W_d^{\mathcal{D}} + W_t^{\mathcal{T}} + W_r^{\mathcal{R}} + W_o^{\mathcal{O}}\right)$. Our factored approach not only allows us to share information between objectives but it also allows us to analyze which stages and primitives are most important to a particular end-task after training is completed (Section 7).

Prescription P2 from Section 4, advocates for introducing as much auxiliary data as possible. As such, instead of fixing to a specific subset throughout training for a particular end-task, we propose to utilize all the objectives in $\mathcal{A}$. This also avoids the combinatorial explosion that comes with exploring subsets of $\mathcal{A}$ at a time. $|\mathcal{A}|$ can be large and descending on all of $\mathcal{A}$ at once can be computationally prohibitive. As an efficient work around, at each training step, we sample a subset of $\mathcal{A}$ for execution with META-TARTAN. Our samples are drawn from all of $\mathcal{A}$ so any objective can get used at any timestep. Because we model each $w^k$ via a factored approach, even if an objective is not sampled its weight is implicitly updated. Our approach is reminiscent of stochastic-relaxation weight sharing (Pham et al., 2018; Dong & Yang, 2019; Li et al., 2020) where sampled architectural primitives result in updates to shared model weights which can be used by other primitives that are not sampled.

We coalesce all the ideas we have introduced so far into Algorithm 1 which we dub **AANG** (**A**utomated **A**uxiliary Learni**NG**). At a high-level, given an end-task **E**:

1. We generate a space of auxiliary objectives $\mathcal{A}$ by leveraging the taxonomy discussed in Section 3. $\mathcal{A}$ may contain auxiliary tasks that can improve our performance on **E**.
2. We leverage MAML-style (Finn et al., 2017) meta-learning to adaptively weight the objectives in $\mathcal{A}$ based on measuring each objective's influence on **E**'s validation set loss.
3. We make our algorithm scalable by sub-sampling the tasks $\mathcal{A}$. By exploiting the underlying structure of the objectives in $\mathcal{A}$ via a factored approach to modeling task weights, we reduce the impact of the inexact sub-sampling.

## 6 EXPERIMENTAL SETTING

Our exploration of auxiliary learning has made the following transitions from the status-quo: manual to automated, single task to multitask, end-task agnostic to end-task aware. In this section, we set up experiments to validate these deviations from the standard.

We focus on continued pre-training (Gururangan et al., 2020; Aghajanyan et al., 2021). In this setting, we perform further auxiliary learning on an already pre-trained model. We favor this setting over pre-training from scratch (Liu et al., 2019b; Yang et al., 2019) not only because it is a more computationally feasible arena for experimentation but also because it is more relevant to modern ML systems where building upon pre-trained models is the norm (Qiu et al., 2020; Du et al., 2020). **Model Details and Datasets:** We use a pre-trained RoBERTa$_{\mathrm{base}}$ (Liu et al., 2019b) as the shared model base. We implement each auxiliary objective as a separate head on top of this shared base. For classification based objectives, the output head is a 2-layer multi-layer perceptron (MLP) that receives representations for the special classification token [CLS] (Devlin et al., 2018) from RoBERTa$_{\mathrm{base}}$. For sequence generation objectives, we make a copy of the pre-trained output layer of RoBERTa$_{\mathrm{base}}$ for each task. Table 4 in Appendix C provides details of the 5 datasets used.

All datasets are low-resource classification tasks. Not only are these datasets more amenable to meta-learning from a computational standpoint, but low-resource tasks also benefit the most from auxiliary learning. We also choose these tasks because they feature in previous work which we use as baselines (Gururangan et al., 2020; Dery et al., 2021b)

**Baselines and Search Spaces:** The following methods are end-task agnostic baselines. By end-task agnostic, we mean that these do not multitask with the end-task. Finetuning on the end-task occurs *after* training on the auxiliary objective.

1. **RoBERTa (Liu et al., 2019b):** We simply finetune a pre-trained RoBERTa$_{base}$ on the end-task.
2. **TAPT (Gururangan et al., 2020):** Continue training RoBERTa$_{base}$ on masked language modelling on end-task data itself before finetuning on the end-task.

The following named objectives are end-task aware baselines that use META-TARTAN (Dery et al., 2021b) but utilize only 1 auxiliary task. Each auxiliary objective is multi-tasked with the end-task.

1. **GPT-style:** We perform end-task aware training with a denoising auxiliary objective based on left-to-right causal masking for computing representations. $\{\mathcal{I}$ = End-task data, $\mathcal{T}$ = No-op, $\mathcal{R}$ = Left-To-Right, $\mathcal{O}$ = Denoise Token $\}$.
2. **XLNET-style:** This is a denoising auxiliary objective that uses randomized masking for computing representations. $\{\mathcal{I}$ = End-task data, $\mathcal{T}$ = No-op, $\mathcal{R}$ = Random-factorized, $\mathcal{O}$ = Denoise Token$\}$.
3. **BERT-style / TAPT:** Denoising inputs corrupted via *BERT-Op*: 80% masking and 10% random replacement. $\{\mathcal{I}$ = End-task data, $\mathcal{T}$ = *BERT-Op*, $\mathcal{R}$ = Bi-directional, $\mathcal{O}$ = Denoise Token$\}$. Please note that this baseline is equivalent to META-TARTAN as introduced in Dery et al. (2021b).

Table 1 details the search spaces that we evaluate against the above baselines. This is by no means the most encompassing search space but we leave more expansive space design to future work. Please note that all tasks within **AANG**-TD, and those with $\{\mathcal{I}$ = End-task$\}$ in **AANG**-TD+ED, are instantiations of task augmentation as introduced in Section 3.

Table 1: **AANG**-TD (task data) has 24 objectives and is based on only end-task data. **AANG**-TD+ED (task data + external data) has 40 objectives and uses both end-task and in-domain data.

|  | $\mathcal{I}$ | $\mathcal{T}$ | $\mathcal{R}$ | $\mathcal{O}$ |
|---|---|---|---|---|
| **TD** | End-task | *BERT-op* Mask | Bi-directional Left-to-Right | Denoise Token End-task |
| **TD+ED** | End-task In-Domain data | Replace No-op | Right-to-Left Random-Factorized | |

**Training Details :** Please see Appendix D for more details about hyper-parameter configurations.

# 7 RESULTS AND DISCUSSION

In this section, we experimentally validate our case for automating the creation of auxiliary objectives and using them in an end-task aware multitask fashion.

## 7.1 GOING A LONG WAY WITHOUT EXTERNAL DATA

We first consider the setting where we rely solely on end-task data (task augmentation), and work with the **AANG**-TD search space. This search space has 24 objectives. Table 2 shows that automatically generating auxiliary objectives from only task data and using them appropriately is productive.

**End-task awareness is key:** From Table 2, methods that are end-task aware result in over $1.12\%$ average improvement over those that are end-task agnostic even under the most generous comparison (GPT-style $79.84\%$ vs task-agnostic TAPT $78.72\%$). Knowing the end-task means that at each iteration, **AANG** can make informed gradient updates by adapting task weights so the resulting auxiliary task better aligns with the end-task (Prescription P1). Amongst the single task objectives, BERT-style performs best. We posit that this is because RoBERTa was trained from scratch on a similar objective and so this objective represents minimal shift in training distributions.

**Adaptive multi-task auxiliary learning improves performance:** We compare single-task end-task aware auxiliary learning to its multitask variant. Table 2 shows that multitasking our 3 different types of language modelling tasks results in improved average performance over using the tasks individually (81.12% for the BERT-style and 81.55% for combining the three single task objectives). We get our best performance when we multitask 24 auxiliary objectives automatically generated with our framework using **AANG**-TD. Boosting the number of objectives from 3 to 24 resulted in a 0.66% improvement in average performance across tasks. This is in line with Prescription P2 from Section 4 since we are increasing the effective amount of auxiliary data. We further posit that introducing more auxiliary objectives also serves to implicitly regularize the end-task during training.

Table 2: Our framework and **AANG** on tasks **using only task data**. Without using any external data, we are able to get significant average performance improvement over baselines. Superscripts are p-values from paired t-tests (best multitask versus best single-task).

| Task Adaptive | Method | # | CS | | BIOMED | NEWS | STANCE | |
|---|---|---|---|---|---|---|---|---|
| | | | ACL-ARC | SCIERC | CHEMPROT | H.PARTISAN | SE-2016-6 | AVG |
| No | RoBERTa | 1 | $66.03_{3.55}$ | $77.96_{2.96}$ | $82.10_{0.98}$ | $93.39_{2.26}$ | $70.37_{1.51}$ | 77.97 |
| | TAPT | 1 | $67.74_{3.68}$ | $79.53_{1.93}$ | $82.17_{0.65}$ | $93.42_{2.87}$ | $70.74_{1.21}$ | 78.72 |
| | [OURS] Static Multitask-TD | 24 | $69.60_{3.80}$ | $\mathbf{83.37}_{0.58}$ | $83.42_{0.26}$ | $97.95_{0.73}$ | $71.02_{0.43}$ | 81.07 |
| Yes | X. GPT-style | 1 | $67.22_{0.44}$ | $81.62_{0.84}$ | $83.29_{1.21}$ | $96.41_{0.73}$ | $70.67_{1.46}$ | 79.84 |
| | Y. XLNET-style | 1 | $69.76_{2.42}$ | $81.81_{0.42}$ | $83.39_{0.31}$ | $96.41_{1.92}$ | $71.18_{0.58}$ | 80.51 |
| | Z. BERT-style (Dery et al., 2021b) | 1 | $70.08_{4.70}$ | $81.48_{0.82}$ | $\mathbf{84.49}_{0.50}^{(0.09)}$ | $96.84_{1.72}$ | $72.70_{0.60}$ | 81.12 |
| | [OURS] AANG-[X+Y+Z] | 3 | $71.51_{3.19}$ | $82.89_{0.78}$ | $83.68_{0.45}$ | $96.92_{1.26}$ | $\mathbf{72.75}_{0.82}^{(0.94)}$ | 81.55 |
| | [OURS] AANG-TD | 24 | $\mathbf{73.26}_{1.32}^{(0.28)}$ | $\mathbf{82.98}_{1.52}^{(0.27)}$ | $83.91_{0.32}$ | $\mathbf{98.46}_{0.0}^{(0.14)}$ | $72.46_{1.65}$ | $\mathbf{82.21}$ |

## 7.2 Introducing External Data

For the ACL-ARC task, we experiment with introducing auxiliary tasks based on external data. **AANG**-TD+ED has 40 tasks, 16 of which are based on domain data. We introduce CS domain data (from the S2ORC dataset (Lo et al., 2019)) that is $n = 10\times$ the size of the task data. From Figure 3 we see that **AANG**-TD+ED makes better use of domain-data than doing end-task aware training using only BERT-style objective with task (TAPT) and domain-data (DAPT) jointly as in Dery et al. (2021b). However, **AANG**-TD+ED (73.70) does not significantly improve over **AANG**-TD (73.26) on the ACL-ARC task (Figure 3). This might seem at odds with Prescription P2 since the TD+ED search space introduces more data. However, note that the AANG search algorithm is approximate and as such, with a larger search space, it can be harder to find composite tasks with a small $\Delta$ as suggested by Prescription P1. We posit that we need more external data than $n = 10\times$ in order to see marked improvements to offset our inexact search of the space of composite functions. However, such scales are outside our computational budget.

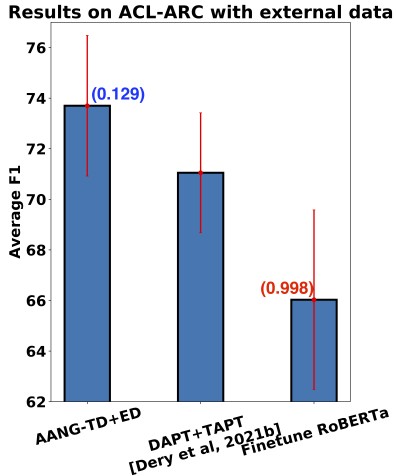

**Results on ACL-ARC with external data**

Figure 3: **AANG** effectively leverages out-of-task data. P-values (in brackets) are comparisons to (Dery et al., 2021b)

## 7.3 Why does AANG Work?

To better understand why our auxiliary learning pipeline improves end-task performance, we perform multiple ablations under **AANG**-TD.
**Static versus Dynamic Weighting:** We ablate the impact of using static task weights throughout training, as against adaptive task weights. Just as with **AANG**, we sub-sample $n$ tasks from the search space at every iteration ($n$ is cross-validated exactly as **AANG** is – Table D ). Each sampled tasks weight is initialized to $\frac{1}{n}$ and this remains unchanged throughout training. This is the Static Multitask-TD baseline in Table 2. **AANG**-TD improves upon the static multitask baseline by over 1.1% on average. With adaptive weighting, **AANG** down-weights objectives that are harmful to the end-task whilst up-weighting relevant ones (Prescription P1). However, using static weightings is more compute friendly since we do not have to calculate task-weight meta-gradients. This compute-vs-performance trade-off is left for practitioners to resolve based on their available resources.
**Impact of number of sampled objectives:** Due to computational constraints, **AANG** sub-samples the set of generated objectives. Whilst this sampling can result in approximation error when inferring task weightings, it can also introduce stochasticity which can help regularize the learned model. From Table 3 (Appendix A) we find that for some tasks (ACL-ARC and SCIERC) sampling a larger number of tasks helps. SE-2016-6 and CHEMPROT on the other hand benefit from smaller number of sampled tasks. Our recommendation is that the number of sampled tasks be cross-validated on a per-task basis.
**Learned task weight trajectories: AANG** learns interesting trajectories for weighting design stage primitives. From Table 2, the fact that **AANG**-TD roughly matches the best single task performance ($72.46_{1.65}$ versus $72.70_{0.60}$ for BERT-style) on the SE-2016-6 task suggests that it may be learning to mostly up-weight this task. Figure 4 provides evidence of this. For the SE-2016-6 task (row 1), composing the highest weighted primitive from each stage [BERT ∘ None ∘ DENOISE] results in BERT-style, the best single task objective. Figure 4 also shows that **AANG** can adapt to overfitting.

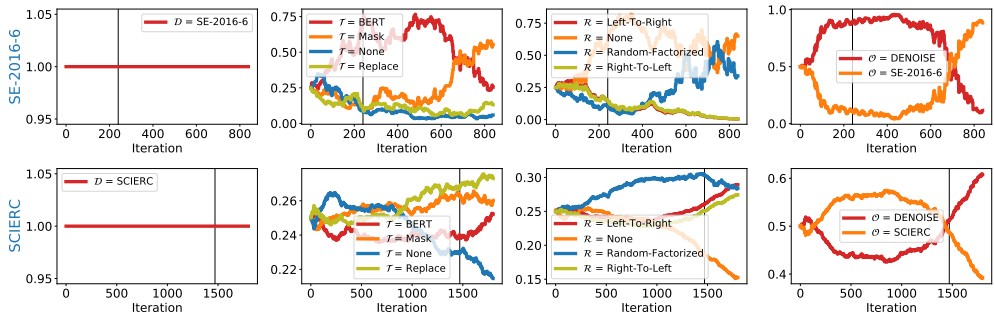

Figure 4: Learned trajectories for **AANG**-TD for run instances of SE-2016-6 and SCIERC tasks.

The vertical black lines indicate the point of best validation set performance. **AANG** responds to over-fitting by down-weighting objectives based on the output loss being over-fit to. Thus, after several iterations, the objective that dominates when the validation performance is at its highest (black vertical line) gets down-weighted in response to it becoming saturated.

**What tasks are important and when they are important?** We study which tasks are most highly weighted early in training (first 10% of learning trajectory) and later in training (last 50%). We aggregate statistics across 3 datasets. Note that early in training, objectives based on the self-

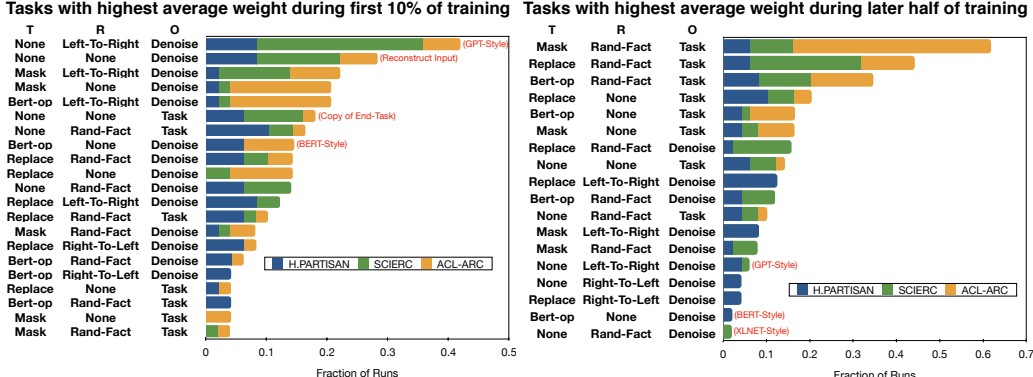

Figure 5: Top ranked objectives (averaged weight) early in training (left) and later in training (right)

supervised output $\mathcal{O} = \{\text{DENOISE}\}$ are highly weighted but later, objectives based on supervised signal, $\mathcal{O} = \{\text{Task}\}$ play a larger role. **AANG** rediscovers the common practice of training on self-supervised objectives before introducing supervised ones. It is also interesting to note that many newly generated objectives (outside of the 3 named single task baselines in Table 2) such as simple input reconstruction were discovered to have relevant impact on the end-tasks. This means **AANG** can automatically surface new, previously unexplored objectives relevant to the end-task.

## 8 LIMITATIONS AND CONCLUSION

Our work has some limitations that we leave for future work. First, because **AANG** relies on meta-learning, it presents extra compute burden over simple multitasking. This is because, we have to independently compute meta-gradients for each auxiliary task thus requiring $\mathcal{O}(n)$ forward-backward operations for $n$ sampled tasks compared to $\mathcal{O}(1)$ for static multitasking. In Table 2, we show that our static Multitask-TD method outperforms all other non-task-adaptive methods by $\approx 2.4\%$ and is thus a viable alternative when runtime is a signficant constraint. Secondly, **AANG** as presented is an approximate algorithm – primarily due to sub-sampling the space of tasks. Thus as mentioned in Section 7.2, we do not get as much gain as desired when our search space becomes larger. We leave finding an efficient exact search algorithm for future exploration.

This paper presents a procedure for automating the creation of auxiliary objectives. We showed, theoretically, how auxiliary learning impacts end-task generalization. This resulted in prescriptions that informed the design of **AANG**, an algorithm to search the space of generated objectives in an end-task aware multitask fashion. Our experiments show that **AANG** is a promising first step in automating auxiliary learning.

# 9 ACKNOWLEDGEMENTS

This work was supported in part by DSO National Laboratories, an ENS-CFM Data Science Chair, DARPA FA875017C0141, the National Science Foundation grants IIS1705121, IIS1838017, IIS2046613 and IIS-2112471, an Amazon Web Services Award, a Facebook Faculty Research Award, funding from Booz Allen Hamilton Inc., and a Block Center Grant. Any opinions, findings and conclusions or recommendations expressed in this material are those of the author(s) and do not necessarily reflect the views of any of these funding agencies. We are grateful for helpful feedback from Uri Alon, Patrick Fernandes, Joon Sik Kim, Han Guo, Victor Akinwande and Clara Na.

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

## A    MORE ABLATION TABLES

Table 3: Varying number of sampled objectives per-iteration.

| Task | $\frac{3}{24}$ tasks | $\frac{6}{24}$ tasks |
|------|------|------|
| ACL-ARC | $72.11_{2.12}$ | $\mathbf{73.26}_{1.32}$ |
| SCIERC | $82.35_{1.76}$ | $\mathbf{82.98}_{1.52}$ |
| SE-2016-6 | $\mathbf{72.46}_{1.65}$ | $\mathbf{72.46}_{0.90}$ |
| CHEMPROT | $\mathbf{83.91}_{0.32}$ | $83.69_{0.98}$ |
| H.PARTISAN | $\mathbf{98.46}_{0.0}$ | $97.95_{0.73}$ |

## B    DISCUSSION OF META-TARTAN (DERY ET AL., 2021B)

META-TARTAN (Dery et al., 2021b) is a MAML style (Finn et al., 2017) meta-learning algorithm that learns to adaptively weight a given set of tasks based on their influence on the end-task validation performance. META-TARTAN achieves this by formulating the following bi-level optimization problem :

$$\theta^*, \mathbf{w}^* = \mathrm{argmin}_{\{\theta \, \in \, g(\theta_0), \, \mathbf{w}\}} \, \mathcal{L}_{\mathbf{E}}(\theta) \tag{2}$$

where

$$\theta_0 = \mathrm{argmin}_\theta \, \mathcal{L}_{\mathrm{total}}(\theta, \mathbf{w}) = \mathrm{argmin}_\theta \, \left( w^* \mathcal{L}_{\mathbf{E}}(\theta) \, + \sum_{T_i \in \mathcal{A}} w_i \mathcal{L}_{T_i}(\theta) \right) \tag{3}$$

Note that $\mathbf{E}$ is the end-task and $\mathcal{A}$ is the set of auxiliary tasks.

Since the above bi-level problem is difficult to solve directly, Dery et al. (2021a) relax the problem and into an alternating optimization problem where task weights are updated based on 1-step improvement to the validation performance of the end-task :

$$\frac{\partial \mathcal{L}_{\mathbf{E}}^{val}(\theta_{t+1}(\mathbf{w}))}{\partial w_i} \approx -\beta \big( \nabla \mathcal{L}_{T_i} \big)^T \big( \nabla \mathcal{L}_{\mathbf{E}}^{val}(\theta_t) \big) \tag{4}$$

To prevent the above relaxation from finding the trivial solution of just upweigting solely the end-task, Dery et al. (2021b) introduce a special dev-head which they use for estimating the meta-gradient :

$$\frac{\partial \mathcal{L}_{T^*}^{val}(\theta^*(\mathbf{w}))}{\partial w_i} \approx -\beta \big( \nabla_\theta \mathcal{L}_{T_i} \big)^T \big( \nabla_\theta \mathcal{L}_{\mathbf{E}}^{val}([\theta_{\mathrm{body}}; \phi^*]_t) \big) \tag{5}$$

Where $\phi_t^*$ is the special dev-head and $\theta_{\mathrm{body}}$ is the body of the model. For even more details about META-TARTAN, please see Section 3 of Dery et al. (2021b).

Though we leverage MET-TARTAN, compared to Dery et al. (2021b), we make three distinct contributions to the field of auxiliary learning. We list them below

1. **Novel Problem Formulation**: As far as we are aware of, we are the first to formulate the problem of automated auxiliary learning. Specifically, we presented an approach for automatically constructing a suite of auxiliary objectives based on existing objectives. Please note that Dery et al. (2021b) perform auxiliary learning with only the DAPT/TAPT variants of the BERT objective. They effectively assume that the search space of objectives (the 2 they explore) is given before-hand. Our approach automatically creates the search space.

2. **Theoretical Novelty**: To the best of our knowledge, we are the first work to provide an exploration of why auxiliary learning improves primary task performance via algorithmic stability. Dery et al. (2021b) in introducing META-TARTAN do not attempt to give a theoretical characterization of why the algorithm improves end-task performance.

3. **Algorithm Improvements to META-TARTAN**: Please note that META-TARAN as presented in Dery et al. (2021b) was used with only 2 auxiliary tasks. When scaling to more tasks, using META-TARTAN naively becomes computationally prohibitive. Specifically, on a search space of N tasks, META-TARTAN requires $O(N)$ order computation per step.

We improve upon this by introducing the task sub-sampling of ($k \ll N$) which reduces the compute overhead to $O(k)$. To account for the impact of sub-sampling as an approximation, we introduced the factorised modelling of task weights which allows sharing of information between auxiliary tasks that might themselves be related.

## C  DATASET DETAILS

Table 4: Specifications of datasets used to evaluate our methods.

| Domain | Task | Label Type | Train Size | Dev Size | Test Size | Classes | Metric |
|---|---|---|---|---|---|---|---|
| BIOMED | CHEMPROT Kringelum et al. (2016) | relation classification | 4169 | 2427 | 3469 | 13 | Accuracy |
| CS | SCIERC Luan et al. (2018) | relation classification | 3219 | 455 | 974 | 7 | F1 |
| STANCE | SE-2016-6 Mohammad et al. (2016) | stance detection | 2497 | 417 | 1249 | 3 | Accuracy |
| CS | ACL-ARC Jurgens et al. (2018) | citation intent | 1688 | 114 | 139 | 6 | F1 |
| NEWS | H.PARTISAN Kiesel et al. (2019) | partisanship | 515 | 65 | 65 | 2 | Accuracy |

## D  MORE TRAINING DETAILS

We run each hyper-parameter configuration across 3 seeds $\{0, 1, 2\}$. We use a batch size of 128 for all end-tasks tasks except H.PARTISAN where we use a batch size of 64. The auxiliary task batch-size, `aux_bsz`, is shared across all the $n$ sub-sampled auxiliary objectives according to the objective's weight.

We use the AdamW optimizer (Loshchilov & Hutter, 2017), with weight decay of 0.01 for all experiments.

Table 5: **AANG**-TD specific Hyper-parameters

| Hyper-parameter | Values | Description |
|---|---|---|
| aux_lr | 1.0, 0.1 | Learning rate for factor vectors - $\{W^{\text{All}}, W^{\mathcal{I}}, W^{\mathcal{T}}, W^{\mathcal{R}}, W^{\mathcal{O}}\}$ |
| sopt_lr | 0.1, 0.01 | Learning rate for primary task weighting $\lambda_e$ |
| nconf_subsamp | 3, 6 | Number of sub-sampled auxiliary tasks. |
| learning rate | 1e-3, 1e-4 | Learning rate used for further training of RoBERTa$_{\text{base}}$ |
| aux_bsz | 256 | Batch size of for auxiliary objectives |

Table 6: **AANG**-TD+ED specific Hyper-parameters

| Hyper-parameter | Values | Description |
|---|---|---|
| aux_lr | 1.0, 0.5, 0.1 | Learning rate for factor vectors - $\{W^{\text{All}}, W^{\mathcal{I}}, W^{\mathcal{T}}, W^{\mathcal{R}}, W^{\mathcal{O}}\}$ |
| sopt_lr | 0.1 | Learning rate for primary task weighting $\lambda_e$ |
| nconf_subsamp | 6, 12, 24 | Number of sub-sampled auxiliary tasks. |
| learning rate | 1e-4 | Learning rate used for further training of RoBERTa$_{\text{base}}$ |
| aux_bsz | 1024 | Batch size of for auxiliary objectives |

Table 7: META-TARTAN Hyper-parameters for single task auxiliary tasks

| Hyper-parameter | Values | Description |
|---|---|---|
| sopt_lr | 1.0, 0.1, 0.01 | Learning rate for primary task weighting $\lambda_e$ |
| learning rate | 1e-3, 1e-4, 5e-5 | Learning rate used for further training of RoBERTa$_{\text{base}}$ |

META-TARTAN introduces a dev-head which is trained sporadically during training for estimating the meta-gradients. We use the following hyper-parameters for training this dev-head : we sample 32 examples (8 examples in the case of H.PARTISAN) and perform full batch gradient descent with

a learning rate of 1e-2 for 10 iterations. The dev-head is trained with the AdamW optimizer with weight decay set to 0.1.

We copy the end-task agnostic baseline results from (Dery et al., 2021b) when available. We use the hyper-parameters specified for TAPT in Gururangan et al. (2020) to train for the SE-2016-6 task.

All models were trained on one of two types of gpus: NVIDIA A100 or NVIDIA A6000. All models fit within a single gpu. We used gradient accumulation to expand the effective batch sizes used for our experiments.

## E   GENERALIZATION ERROR BOUND FOR END-TASK AWARE TRAINING

### E.1   DEFINITIONS

**Definition E.1.** *A function, $f : \Omega \to \mathbb{R}$ is L-Lipschitz if $\forall u, v \in \mathrm{dom}(f)$:*

$$\|f(u) - f(v)\| \le L\|u - v\|$$

*Note that L-Lipschitz implies bounded gradients.*

$$\|\nabla f(w)\| \le L \quad \forall w$$

**Definition E.2.** *A function, $f : \Omega \to \mathbb{R}$ is $\beta$-smooth if $\forall u, v \in \Omega$:*

$$\|\nabla f(u) - \nabla f(v)\| \le \beta\|u - v\|$$

**Definition E.3.** *An update rule, G is $\sigma$-bounded if :*

$$\sup_{w \in \Omega} \|w - G(w)\| \le \sigma$$

Consider the following general setting. There is an unknown distribution $\mathcal{D}_e$ over examples from some space $\mathcal{Z}$. We receive a sample $S = (z_1, \ldots, z_{N_e})$ of $N_e$ examples drawn i.i.d. from $\mathcal{D}_e$. Our goal is to find a model $w$, that parameterizes the function $f_e$, with small population risk defined as:

**Definition E.4.** *Population Risk*

$$R[w] = \mathbf{E}_{z \sim \mathcal{D}_e} f_e(w; z)$$

**Definition E.5.** *Empirical Risk*
*Since we have a finite number of samples, we can only compute the empirical risk which is :*

$$R_S[w] = \frac{1}{N_e} \sum_i f_e(w; z_i),$$

Let $A$ be a potentially randomized algorithm (such as Stochastic Gradient Descent) that is a function of the $S$ such that $w = A(S)$.

**Definition E.6.** *Generalization Error $\epsilon_{gen}(A, N_e)$*

$$\epsilon_{gen}(A, N_e) = \mathbf{E}_{S,A}\big[R_S[A(S)] - R[A(S)]\big]$$

**Definition E.7.** *Uniform Stability*
*A randomized algorithm A is $\epsilon$-uniformly stable if for all data sets $S, S' \in \mathcal{Z}$, $|S| = |S'| = N_e$ such that $S$ and $S'$ differ in at most one example, we have*

$$\sup_z \mathbf{E}_A\big[f_e(A(S); z) - f_e(A(S'); z)\big] \le \epsilon$$

*Here, the expectation is taken only over the internal randomness of A. We will denote by $\epsilon_{\mathrm{stab}}(A, N_e)$ the infimum over all $\epsilon$ for which the above holds.*

### E.2   RELEVANT THEOREMS

**Theorem E.1** (Uniform Stability implies Generalization in expectation)**.** *Let Algorithm A be $\epsilon$-uniformly stable. Then,*

$$\epsilon_{gen}(A, N_e) = \left|\mathbf{E}_{S,A}\big[R_S[A(S)] - R[A(S)]\big]\right| \le \epsilon_{stab}(A, N_e)$$

*For full proof see Theorem 2.2 of Hardt et al. (2016).*

**Theorem E.2** (Stochastic Gradient Method is stable). *Assume that $f_e(\cdot; z) \in [0, 1]$ is an L-Lipschitz and $\beta_e$-smooth loss function for every $z$. Suppose that we run SGM for $T$ steps with monotonically non-increasing step sizes $\alpha_t \leq \frac{c}{t}$. Then, SGM has uniform stability with :*

$$\epsilon_{sgm} \leq \frac{1 + \frac{1}{q}}{N_e - 1} \left(2cL^2\right)^{\frac{1}{q+1}} T^{\frac{q}{q+1}}$$

$$\text{where } q = \beta_e c$$

*We can simplify this to only terms involving $T$ and $N_e$*

$$\epsilon_{sgm} \lessapprox \frac{T^{1 - \frac{1}{c\beta_e + 1}}}{N_e} \tag{6}$$

*Proof.* For the full proof, see *Theorem 3.12* of Hardt et al. (2016)

$\square$

### E.3 GROWTH FUNCTIONS

**Lemma E.3** (Growth Recursion Under Dynamic Sampling). *We consider the Stochastic Gradient update rule $G : \Omega \to \Omega$ :*

$$G_f(w) = w - \alpha \nabla f(w)$$

*Fix an arbitrary sequence of updates $G_{f_1}, \ldots, G_{f_T}$ and another $G'_{f_1}, \ldots, G'_{f_T}$. Let $w_0 = w'_0$ be a starting point in $\Omega$ given that $f : \Omega \to \mathbb{R}$ and define*

$$\delta_t = \mathbb{E}_{f_1 \ldots f_t \sim \mathcal{P}_\lambda} \left[\|w_t - w'_t\|\right]$$

*where $w_t, w'_t$ are defined recursively through :*

$$w_t = G_{f_t}(w_{t-1}) \quad w'_t = G'_{f_t}(w'_{t-1}) \quad t \geq 0$$

*Then we have the recurrence relation :*

$$\delta_0 = 0$$
$$\delta_{t+1} \leq \begin{cases} \min\left\{\left(1 + \alpha\lambda_1\beta_1\right)\delta_t + \alpha\lambda_2\left(\Delta + 2L\right), \ \left(1 + \alpha\left(\lambda_1\beta_1 + \lambda_2\beta_2\right)\right)\delta_t\right\} & G_{f_t} = G'_{f_t} \\ \delta_t + 2\sigma_t & G_{f_t}, G'_{f_t} \text{ are } \sigma\text{-bounded} \end{cases}$$

*Note that $\mathcal{P}_f$ is a distribution over the support $\{f^1, f^2\}$ according to probabilities $\{\lambda_1, \lambda_2 \mid \lambda_1 + \lambda_2 = 1\}$. $\{f_1, f_2\}$ have smoothness $\beta_1, \beta_2$ respectively.*

*Proof.* The second bound on $\delta_t$ is taken directly from Lemma 2.5 of Hardt et al. (2016). We now derive the first-half of the first bound

$$\delta_{t+1} = \mathbb{E}_{f_1\ldots f_{t+1}\sim\mathcal{P}_\lambda}\big[\|w_{t+1} - w'_{t+1}\|\big]$$

$$= \mathbb{E}_{f_1\ldots f_t\sim\mathcal{P}_\lambda}\Big[\lambda_1\|G_{f^1}(w_t) - G'_{f^1}(w'_t)\| + \lambda_2\|G_{f^2}(w_t) - G'_{f^2}(w'_t)\|\Big]$$

$$= \mathbb{E}_{f_1\ldots f_t\sim\mathcal{P}_\lambda}\Big[\lambda_1\|w_t - \alpha\nabla f^1(w_t) - w'_t + \alpha\nabla f^1(w'_t)\| + \lambda_2\|w_t - \alpha\nabla f^2(w_t) - w'_t + \alpha\nabla f^2(w'_t)\|\Big]$$

$$\leq \mathbb{E}_{f_1\ldots f_t\sim\mathcal{P}_\lambda}\big[\|w_t - w'_t\|\big] + \alpha\mathbb{E}_{f_1\ldots f_t\sim\mathcal{P}_\lambda}\Big(\lambda_1\|\nabla f^1(w'_t) - \nabla f^1(w_t)\| + \lambda_2\|\nabla f^2(w'_t) - \nabla f^2(w_t)\|\Big)$$

(Triangle Inequality used for above step)

$$= \delta_t + \alpha\mathbb{E}_{f_1\ldots f_t\sim\mathcal{P}_\lambda}\Big(\lambda_1\|\nabla f^1(w'_t) - \nabla f^1(w_t)\| + \lambda_2\|\nabla f^2(w'_t) - \nabla f^2(w_t)\|\Big)$$

(Without Loss of Generality, let $\beta_1 \leq \beta_2$)

$$\leq \delta_t + \alpha\mathbb{E}_{f_1\ldots f_t\sim\mathcal{P}_\lambda}\Big[\lambda_1\beta_1\|w_t - w'_t\| + \lambda_2\|\nabla f^2(w'_t) - \nabla f^2(w_t)\|\Big] \quad \text{(Smoothness)}$$

$$= \delta_t + \alpha\lambda_1\beta_1\delta_t + \alpha\lambda_2\mathbb{E}_{f_1\ldots f_t\sim\mathcal{P}_\lambda}\Big[\|\nabla f^2(w'_t) - \nabla f^2(w_t)\|\Big] \quad \text{(Triangle Inequality)}$$

$$= (1 + \alpha\lambda_1\beta_1)\delta_t + \alpha\lambda_2\Big\|\nabla f^2(w'_t) - \nabla f^1(w'_t) + \nabla f^1(w'_t) - \nabla f^2(w_t)\Big\| \quad \text{(add zero)}$$

$$\leq (1 + \alpha\lambda_1\beta_1)\delta_t + \alpha\lambda_2\Big(\|\nabla f^2(w'_t) - \nabla f^1(w'_t)\| + \|\nabla f^1(w'_t) - \nabla f^2(w_t)\|\Big) \quad \text{(Triangle Inequality)}$$

$$\leq (1 + \alpha\lambda_1\beta_1)\delta_t + \alpha\lambda_2\Big(\Delta + \|\nabla f_1(w'_t) - \nabla f_2(w_t)\|\Big) \quad \text{Using Assumption A.1}$$

$$\leq (1 + \alpha\lambda_1\beta_1)\delta_t + \alpha\lambda_2\Big(\Delta + \|\nabla f_1(w'_t)\| + \|\nabla f_2(w_t)\|\Big) \quad \text{Triangle Inequality}$$

$$\leq (1 + \alpha\lambda_1\beta_1)\delta_t + \alpha\lambda_2\big(\Delta + 2L\big) \quad \text{$L$-Lipschitz function}$$

To obtain the second half of the first bound :

$$\delta_{t+1} = \mathbb{E}_{f_1\ldots f_{t+1}\sim\mathcal{P}_\lambda}\big[\|w_{t+1} - w'_{t+1}\|\big]$$

$$= \mathbb{E}_{f_1\ldots f_t\sim\mathcal{P}_\lambda}\Big[\lambda_1\|G_{f^1}(w_t) - G'_{f^1}(w'_t)\| + \lambda_2\|G_{f^2}(w_t) - G'_{f^2}(w'_t)\|\Big]$$

$$= \mathbb{E}_{f_1\ldots f_t\sim\mathcal{P}_\lambda}\Big[\lambda_1\|w_t - \alpha\nabla f^1(w_t) - w'_t + \alpha\nabla f^1(w'_t)\| + \lambda_2\|w_t - \alpha\nabla f^2(w_t) - w'_t + \alpha\nabla f^2(w'_t)\|\Big]$$

$$\leq \mathbb{E}_{f_1\ldots f_t\sim\mathcal{P}_\lambda}\big[\|w_t - w'_t\|\big] + \alpha\mathbb{E}_{f_1\ldots f_t\sim\mathcal{P}_\lambda}\Big(\lambda_1\|\nabla f^1(w'_t) - \nabla f^1(w_t)\| + \lambda_2\|\nabla f^2(w'_t) - \nabla f^2(w_t)\|\Big)$$

(Triangle Inequality used for above step)

$$\leq \delta_t + \alpha\mathbb{E}_{f_1\ldots f_t\sim\mathcal{P}_\lambda}\Big[\lambda_1\beta_1\|w_t - w'_t\| + \lambda_2\beta_2\|w_t - w'_t\|\Big] \quad \text{(Smoothness)}$$

$$= \delta_t + \alpha\lambda_1\beta_1\mathbb{E}_{f_1\ldots f_t\sim\mathcal{P}_\lambda}\Big[\|w_t - w'_t\|\Big] + \alpha\lambda_2\beta_2\mathbb{E}_{f_1\ldots f_t\sim\mathcal{P}_\lambda}\Big[\|w_t - w'_t\|\Big]$$

$$= \delta_t + \alpha(\lambda_1\beta_1 + \lambda_2\beta_2)\delta_t$$

$$= (1 + \alpha(\lambda_1\beta_1 + \lambda_2\beta_2))\delta_t$$

$$\square$$

### E.4  STABILITY OF DYNAMIC SAMPLING

We repeat the description of our Auxiliary Learning with Dynamic Sampling Setting here for ease of access.

**Setting** : We are given an auxiliary objective $f_a(\cdot; z) \in [0, 1]$ with $N_a$ samples $S_a = (z_1, \ldots, z_{N_a})$ from the distribution $\mathcal{D}_a$. At any iteration of SGD, we sample a choice of either the end-task function $f_e$ or the auxiliary objective $f_a$ according to the probabilities $\lambda_e, \lambda_a \mid \lambda_e + \lambda_a = 1$. Given the chosen objective, we sample a data-point and perform stochastic gradient descent (SGD) based on the sampled data-point.

An equivalent way to instantiate this procedure to create $S_A$ by drawing $N' = N_e + N_a$ total samples from the end-task and auxiliary task according to $\mathcal{P}_\lambda$. $S'_A$ is then created by replacing 1 end-task sample in $S_A$. At each step, a sample is drawn from a distribution : $z_i, z'_i \sim P_{S_A}, P_{S'_A}$ and a gradient step is taken on the function corresponding to the set the sample was drawn from.

**Lemma E.4** (Stability of dynamic sampling). *We denote the outputs of $T$ steps of SGM on $S_A$ and $S'_A$ with the dynamically sampled functions, as $w_T$ and $w'_T$ respectively. Then, for every $z_e \in Z_e$ and every $t_0 > 0$, under both the random update rule and the random permutation rule, we have :*

$$\mathbb{E}\big|f_e(w_T; z) - f_e(w'_T; z)\big| \leq \frac{\gamma t_0}{N'} \sup_{w, z_e} f_e(w; z_e) + L\mathbb{E}[\delta_T | \delta_{t_0} = 0]$$

*Where $N' = N_e + N_a$ and $\gamma = \frac{\lambda_e \cdot N'}{N_e} = \frac{\lambda_e}{\lambda^r}$.*

*Proof.* Let $\mathcal{E} = \mathbf{1}[\delta_{t_0} = 0]$ denote the event that $\delta_{t_0} = 0$. We have

$$
\begin{aligned}
\mathbb{E}\big|f_e(w_T; z) - f_e(w'_T; z)\big| &= P\{\mathcal{E}\}\mathbb{E}\big[\big|f_e(w_T; z) - f_e(w'_T; z)\big| \big| \mathcal{E}\big] \\
&\quad + P\{\mathcal{E}^c\}\mathbb{E}\big[\big|f_e(w_T; z) - f_e(w'_T; z)\big| \big| \mathcal{E}^c\big] \\
&\leq \mathbb{E}\big[\big|f_e(w_T; z) - f_e(w'_T; z)\big| \big| \mathcal{E}\big] + P\{\mathcal{E}^c\} \cdot \sup_{w, z_e} f_e(w; z_e) \\
&\qquad \text{because } f_e \text{ is non-negative} \\
&\leq L\mathbb{E}\big[\|w_T - w'_T\| \big| \mathcal{E}\big] + P\{\mathcal{E}^c\} \cdot \sup_{w, z_e} f_e(w; z_e)
\end{aligned}
\tag{7}
$$

because $f_e$ is $L$-Lipschitz

We now proceed to bound $P\{\mathcal{E}^c\}$. Let $i_* \in [N']$ denote the position in which $S_A, S'_A$ differ and consider the random variable I assuming the index of the first time step in which SGM uses the example $z_e^{i_*}$. Note that when $I > t_0$, then we must have that $\delta_{t_0} = 0$ since the two samples are identical up until this point.

$$P\{\mathcal{E}^c\} = P\{\delta_0 \neq 0\} \leq P\{I \leq t_0\}$$

Using the selection rule specified above (sample either $f_e, f_a$ according to the probabilities $\lambda_e, \lambda_a$ and then sample uniformly from the selected task data) we have that :

$$P\{I \leq t_0\} = \sum_{t=1}^{t_0} P\{I = t_0\} = \sum_{t=1}^{t_0} \left(\lambda_e \cdot \frac{1}{N_e}\right) = \frac{\lambda_e t_0}{N_e} = \frac{\gamma t_0}{N'}$$

$\square$

**Theorem E.5** (Stability Bound on Dynamic Sampling). *Assume that $f_e(; z_e), f_a(; z_a) \in [0, 1]$ are $L$-Lipschitz and $\beta_e$ and $\beta_a$-smooth loss functions. Consider that we have $N' = N_e + N_a$ total samples where $f_e$ and $f_a$ have $N_e$ and $N_a$ samples respectively. Suppose that we run SGM for $T$ steps with monotonically non-increasing step sizes $\alpha_t \leq \frac{c}{t}$ by dynamically sampling the tasks according to $\lambda_e$ and $\lambda_a$. Then, with respect to $f_e$, SGM has uniform stability with :*

$$\epsilon_{\text{stab}} \leq \left(1 + \frac{1}{c\bar{\beta}}\right)\left(\frac{2\gamma L^2 c}{N' - \gamma} + \rho Lc\right)^{\frac{1}{c\bar{\beta}+1}} \left(\frac{\gamma T}{N'}\right)^{\frac{c\bar{\beta}}{1+c\bar{\beta}}}$$

$$\text{Where} \quad \gamma = \frac{\lambda_e N'}{N_e}$$

*Given that $\beta^* = \min\{\beta_e, \beta_a\}$ and $\lambda^*$ is the corresponding weighting of the function with smaller smoothness.*

*Depending on which one gives a tighter bound the pair $(\bar{\beta}, \rho)$ can be :*

$$(\bar{\beta}, \rho)_1 = (\lambda^* \beta^*, \ (1 - \lambda^*)(\Delta + 2L))$$

*or*

$$(\bar{\beta}, \rho)_2 = (\lambda_e \beta_e + \lambda_a \beta_a, \ 0)$$

When $(\bar{\beta}, \rho)_1$ gives the tighter bound, we can simplify to :

$$\epsilon_{\text{gen}} \lesssim (\Delta)^{\frac{1}{1+c\lambda^*\beta^*}} \left(\frac{\gamma T}{N'}\right)^{1-\frac{1}{c\lambda^*\beta^*+1}}$$

As presented in Section 4.

*Proof.* Let $S_A, S'_A$ be two sample of size $N' = N_e + N_a$ as described in lemma E.4. Consider the gradient updates $G_{f_1}, \ldots, G_{f_T}$ and $G'_{f_1}, \ldots, G'_{f_T}$ induced by running SGM on samples $S_A$ and $S'_A$ respectively. Let $w_T$ and $w'_T$ denote the corresponding outputs of SGM. By lemma E.4 we have :

$$\mathbb{E}|f_e(w_T; z) - f_e(w'_T; z)| \leq \frac{\gamma t_0}{N'} \sup_{w, z_e} f_e(w; z_e) + L\mathbb{E}[\delta_T | \delta_{t_0} = 0] \tag{8}$$

Let $\Psi_T = \mathbb{E}[\delta_T | \delta_{t_0} = 0]$. We will bound $\Psi_T$ as function of $t_0$ and then minimize for $t_0$. Note the following :

- At any step $t$, with probability $\left(1 - \frac{\gamma}{N'}\right)$, the sample selected is the same in both $S_A$ and $S'_A$. In this case $G_{f_t} = G'_{f_t}$ and we use the corresponding expansivity rule from lemma E.4. This gives :

$$\delta_{t+1} \leq \min\left\{(1 + \alpha_t\lambda^*\beta^*)\delta_t + \alpha_t(1-\lambda^*)(\Delta + 2L), \ (1 + \alpha_t(\lambda_e\beta_e + \lambda_a\beta_a))\delta_t\right\}$$

  Where $\beta^* = \min\{\beta_e, \beta_a\}$ and $\lambda^*$ is the corresponding weighting of the function with smaller smoothness. To avoid deriving the bound independently for each case, we perform a variable substituation that captures the two cases :

$$\delta_{t+1} \leq (1 + \alpha_t\bar{\beta})\delta_t + \alpha_t\rho$$

  $\bar{\beta} = \{\lambda^*\beta^*, \ \lambda_e\beta_e + \lambda_a\beta_a\}$ and $\rho = \{(1-\lambda^*)(\Delta + 2L), 0\}$. We can present the final bound in terns of these variables which can be substituted depending on the minimizer.

- With probability $\frac{\gamma}{N'}$ the selected example is different. Note that in this case, we know that we are evaluating the end-task function $f_e$. We use that both $G_{f_t}$ and $G'_{f_t}$ are $(\sigma_t = \alpha_t L)$-bounded according to lemma E.3 since $f_e$ is $L$-Lipschitz.

Combining the above we have :

$$\begin{aligned}
\Psi_{t+1} &\leq \left(1 - \frac{\gamma}{N'}\right)\left((1 + \alpha_t\bar{\beta})\Psi_t + \alpha_t\rho\right) + \frac{\gamma}{N'}(\Psi_t + 2\alpha_t L) \\
&= \left(\frac{\gamma}{N'} + \left(1 - \frac{\gamma}{N'}\right)(1 + \alpha_t\bar{\beta})\right)\Psi_t + \frac{2\gamma\alpha_t L}{N'} + \alpha_t\left(1 - \frac{\gamma}{N'}\right)\rho \\
&= \left(1 + \left(1 - \frac{\gamma}{N'}\right)\alpha_t\bar{\beta}\right)\Psi_t + \frac{\alpha_t(2\gamma L + (N' - \gamma)\rho)}{N'} \\
&\leq \left(1 + \left(1 - \frac{\gamma}{N'}\right)\frac{c}{t}\bar{\beta}\right)\Psi_t + \frac{c(2\gamma L + (N' - \gamma)\rho)}{tN'} \\
&\leq \exp\left(\left(1 - \frac{\gamma}{N'}\right)\frac{c}{t}\bar{\beta}\right)\Psi_t + \frac{c(2\gamma L + (N' - \gamma)\rho)}{tN'} \\
&\quad \text{We use } 1 + x \leq \exp(x) \ \forall x \\
&\leq \exp\left(\left(1 - \frac{\gamma}{N'}\right)\frac{c}{t}\bar{\beta}\right)\Psi_t + \frac{c\bar{\rho}}{tN'} \\
&\quad \text{Where } \bar{\rho} = (2\gamma L + (N' - \gamma)\rho)
\end{aligned} \tag{9}$$

We can unwind the recurrence until $\Psi_{t_0} = 0$.

$$\Psi_T \leq \sum_{t=t_0+1}^{T} \left( \prod_{k=t+1}^{T} \exp\left( (1 - \frac{\gamma}{N'}) \frac{c\bar{\beta}}{k} \right) \right) \left( \frac{c\bar{\rho}}{tN'} \right)$$

$$= \sum_{t=t_0+1}^{T} \left( \frac{c\bar{\rho}}{tN'} \right) \exp\left( (1 - \frac{\gamma}{N'}) c\bar{\beta} \sum_{k=t+1}^{T} \frac{1}{k} \right)$$

$$\leq \sum_{t=t_0+1}^{T} \left( \frac{c\bar{\rho}}{tN'} \right) \exp\left( (1 - \frac{\gamma}{N'}) c\bar{\beta} \log\left( \frac{T}{t} \right) \right)$$

$$= \frac{c\bar{\rho} T^{c\bar{\beta}(1-\frac{\gamma}{N'})}}{N'} \sum_{t=t_0+1}^{T} t^{-c\bar{\beta}(1-\frac{\gamma}{N'})-1} \tag{10}$$

We can upper bound the sum over t with an integral + drop negative terms

$$\leq \frac{c\bar{\rho}}{N'c\bar{\beta}(1-\frac{\gamma}{N'})} \left( \frac{T}{t_0} \right)^{c\bar{\beta}(1-\frac{\gamma}{N'})}$$

$$= \frac{\bar{\rho}}{\bar{\beta}(N'-\gamma)} \left( \frac{T}{t_0} \right)^{c\bar{\beta}(1-\frac{\gamma}{N'})}$$

$$\leq \frac{\bar{\rho}}{\bar{\beta}(N'-\gamma)} \left( \frac{T}{t_0} \right)^{c\bar{\beta}}$$

Plugging this bound back into Equation 8 and using the fact that $f_e \in [0,1]$:

$$\mathbb{E}\left| f_e(w_T; z) - f_e(w'_T; z) \right| \leq \frac{\gamma t_0}{N'} + \frac{L\bar{\rho}}{\bar{\beta}(N'-\gamma)} \left( \frac{T}{t_0} \right)^{c\bar{\beta}} \tag{11}$$

We let $q^* = c\bar{\beta}$, we can minimize the R.H.S by setting :

$$t_0 = \left( \frac{N'Lc\bar{\rho}}{\gamma(N'-\gamma)} \right)^{\frac{1}{q^*+1}} T^{\frac{q^*}{q^*+1}}$$

Plugging this in gives us :

$$\mathbb{E}\left| f_e(w_T; z) - f_e(w'_T; z) \right| \leq \left( \frac{(1+\frac{1}{c\bar{\beta}})}{N'} \right) \left( \frac{N'Lc(2\gamma L + (N'-\gamma)\rho)}{(N'-\gamma)} \right)^{\frac{1}{c\bar{\beta}+1}} (\gamma T)^{\frac{c\bar{\beta}}{1+c\bar{\beta}}}$$

$$= \left( 1 + \frac{1}{c\bar{\beta}} \right) \left( \frac{2\gamma L^2 c}{N'-\gamma} + \rho Lc \right)^{\frac{1}{c\bar{\beta}+1}} \left( \frac{\gamma T}{N'} \right)^{\frac{c\bar{\beta}}{1+c\bar{\beta}}} \tag{12}$$

Recall that :

$$\bar{\beta} = \left\{ \lambda^* \beta^*, \ \lambda_e \beta_e + \lambda_a \beta_a \right\}$$

$$\rho = \left\{ (1-\lambda^*)(\Delta + 2L), 0 \right\}$$

We can choose whichever of the pairs for $\bar{\beta}, \rho$ that minimizes the bound : $\qquad \square$

## F    DISCUSSION OF GENERALIZATION ERROR BOUNDS

### F.1    WHAT DOES THEOREM E.5 SAY.

We consider the setting where

$$\bar{\beta} = \lambda^* \beta^*$$

$$\rho = (1-\lambda^*)(\Delta + 2L)$$

Assuming the $\rho$ term dominates Equation 12 in this setting is :

$$
\begin{aligned}
\epsilon_{\text{gen}}^{\text{auxdyn}} \leq \epsilon_{\text{stab}}^{\text{auxdyn}}\big|_{(\bar{\beta},\rho)_1} &\lesssim \sqrt[1+c\bar{\beta}]{(1-\lambda^*)(\Delta+2L)}\left(\frac{\gamma T}{N'}\right)^{\frac{c\bar{\beta}}{1+c\bar{\beta}}} \\
&\lesssim (\Delta)^{\frac{1}{1+c\lambda^*\beta^*}}\left(\frac{\gamma T}{N'}\right)^{1-\frac{1}{c\lambda^*\beta^*+1}} \quad \text{This is Equation 1 from Section 4}
\end{aligned}
$$

$$(13)$$

In going from the first line to the second we consider the setting where $\Delta \gg 2L$. This is a case where the auxiliary task is sufficiently different from the primary task. Some observations about this setting:

1. Smaller $\Delta$ implies auxiliary task is similar to main task and leads to improving the bound.

2. Dependence of the bound on $N'$ is a bit more nuanced. **Note that increasing $N'$ increases $\gamma$ unless we reduce $\lambda_e$ appropriately**. Remember that $\lambda_e$ is the rate at which we sample the primary task. Thus, if we add more auxiliary data but still sample the primary task at the original rate, then we are effectively ignoring the extra auxiliary data.

3. It might be tempting to assume that we can get arbitrary improvements in this setting by setting $\lambda_e = 0$. However, **note that whilst this might reduce the generalization error**, it means that we are seeing none of the end-task which would result in large **increase in the training error**

4. Note that $(\bar{\beta} = \lambda^*\beta^* \leq \beta_e)$ always. So we get improvements on the dependence on $T$ compared to Theorem E.2.

5. We can optimize $\lambda_e, \lambda_a$ to minimize $\epsilon_{\text{stab}}^{\text{auxdyn}}$.

