# OpenReview forum: "AANG : Automating Auxiliary Learning"
_ICLR.cc/2023/Conference — ICLR 2023 notable top 25%_

### Official Review · Reviewer_pGCt · 2022-10-14

**Confidence:** 4
**Clarity, Quality, Novelty And Reproducibility:** The paper is easy to follow and is qu…
**Correctness:** 4
**Technical Novelty And Significance:** 4
**Empirical Novelty And Significance:** 4
**Recommendation:** 8

**Strength And Weaknesses:**

**Strength**
* This is the first work that propose to automatically generate auxilliary objectives
* The experimental resutls seems promising


**Weaknesses**
* The proposed method presents (potentially heavy) extra computational cost
* The proposed method only run experiments on continued pretraining setting and the datasets are relatively small.

**Summary Of The Paper:**

The pretrain-(continued-pretrain)-finetune paradigm has been de-facto in recent NLP state-of-the-arts. However, the construction of auxiliary objectives still need carefully design by experts. This paper proposes to automatically generate a suite of auxiliary objectives. Specifically, they provide theoretical bound for the generalization error in auxiliary learning, and design an efficient algorithm for searching the space of generated objectives to find those most useful to a specified end-task.

**Summary Of The Review:**

To the best of my knowledge, this is the first work that propose to automatically generate auxilliary objectives. Moreover, it provides good results both theoretically and practically.

---

> ### Author Response · Authors · 2022-11-10
> **Response to Reviewer pGCt**
>
> We would like to thank the reviewer for committing time to provide feedback on our paper and appreciate the positive words about the paper being novel and solid. We address specific concerns below :
>
> > ““The proposed method presents (potentially heavy) extra computational cost”
>
> We recognize that using meta-learning comes at an extra computational cost. We acknowledge this fact in Section 8 (Limitations and Conclusion) and presented the Static Multitask-TD variant (Table 2) as a more compute-friendly alternative that still leads to significant improvements over comparable baselines like TAPT and vanilla fine-tuning.
> In the case where the extra computational overhead of meta-learning can be borne, we presented the task sub-sampling approach in Section 5 in order to prevent the meta-learning cost from prohibitively scaling with the size of the search space.
> Please see also our general response for more discussions on the computational cost of AANG
>
> ---
>
> > ““The proposed method only run experiments on continued pretraining setting and the datasets are relatively small”
>
> We acknowledge that our method is general enough that it can be used in the pre-training setting. Unfortunately, we do not have the compute resources to experiment with using AANG with large scale pre-training. This inspired us to focus on the continued pre-training setting. The continued pre-training is very popular amongst everyday practitioners whom we target. Our choice of datasets was informed by the fact that :
> 1. These datasets were featured in previous work on continued pre-training [1, 2] which we feature as baselines.
> 2. Auxiliary learning has been shown to improve results more in the low-resource setting [1, 2, 3]
> 3. Low resource problems are relatively ignored in the era of big-data and large models.
>
> [1] Gururangan, Suchin, et al. "Don't stop pretraining: adapt language models to domains and tasks." arXiv preprint arXiv:2004.10964 (2020)
>
> [2] Dery, Lucio M., et al. "Should we be pre-training? an argument for end-task aware training as an alternative." arXiv preprint arXiv:2109.07437 (2021).
>
> [3] Weighted Training for Cross-Task Learning: https://openreview.net/forum?id=ltM1RMZntpu

---

### Official Review · Reviewer_rBEM · 2022-10-25

**Confidence:** 4
**Clarity, Quality, Novelty And Reproducibility:** What if there is more than one end task?
**Correctness:** 3
**Technical Novelty And Significance:** 4
**Empirical Novelty And Significance:** 4
**Recommendation:** 8

**Strength And Weaknesses:**

This paper proposes AANG to automatically generate a suite of auxiliary objectives for end tasks. AANG builds a unified framework/pipeline that consists of several parts: input data, input transformation, model representation, and output. AANG uses algorithmic stability to search for the best combination in each part. The experiments are conducted on the NLP domain, and the results demonstrate the effectiveness of the proposed method.

Even though many techniques used in this paper are proposed by other works, this paper delicately combines them.

The paper presents a solid theoretical analysis of the proposed algorithm.

It would be better to extend and deepen the experiment to non-NLP tasks.

**Summary Of The Paper:**

This paper proposes AANG to automatically generate a suite of auxiliary objectives for end tasks. AANG builds a unified framework/pipeline that consists of several parts: input data, input transformation, model representation, and output. AANG uses algorithmic stability to search for the best combination in each part. The experiments are conducted on the NLP domain, and the results demonstrate the effectiveness of the proposed method.

**Summary Of The Review:**

I think this paper is novel and soild enough. It can meet the acceptance line of ICLR.

---

> ### Author Response · Authors · 2022-11-10
> **Response to Reviewer rBEM**
>
> Thank you for committing time to provide feedback on our paper and appreciate the recognition that our work is “novel and solid”. We address specific concerns below :
>
> > ““It would be better to extend and deepen the experiment to non-NLP tasks”
>
> We recognize the potential for AANG to be leveraged in many domains, not just NLP.  While the current paper focuses on NLP tasks, we definitely believe that automating auxiliary learning in other domains is a promising future direction of exploration.
>
> ---
>
> > “What if there is more than one end-task”
>
> AANG is easily extensible to the setting of multiple end-tasks. In this case, we can construct the meta-learning objective as a weighted aggregate of the individual end-task losses. The input and output sections of our pipeline would also be updated to reflect that there are now multiple end-task datasets and losses we can choose from. This expands the search space of auxiliary objectives.

---

### Official Review · Reviewer_KWWF · 2022-10-28

**Confidence:** 3
**Correctness:** 3
**Technical Novelty And Significance:** 2
**Empirical Novelty And Significance:** 2
**Recommendation:** 5

**Clarity, Quality, Novelty And Reproducibility:**

The overall clarity of the paper can be improved by providing more details about the training algorithm and a high-level description.

The approach seems novel, though it builds and relies heavily on META-TARTAN. The new thing seems to be about the change in dynamic weight updates and it would be good to clarify these differences.

Code is not provided, though there are more experimental details and hyper-parameters in supplementary to help with reproducibility.

**Strength And Weaknesses:**

Strengths
- An approach to dynamically learn auxiliary loss weighting that could be useful for finetuning pre-trained models on specific tasks.
- Empirical results include statistical significance, and interesting analysis of how weights vary during training.

Weaknesses
- The paper was a bit tough to follow. The setting up of the auxiliary loss space was more detailed than it needed to be. In particular, I do not find this setup to be specially illuminating, or inspiring new losses just by its construction (as one would not trivially expect every possible choice of loss in this space to be useful). It would have been better to provide more details in sections 4/5/6. The paper also assumes knowledge of META-TARTAN paper and heavily relies on it.
- Differences from the META-TARTAN paper are not clear and are not explicitly mentioned. It should be an important baseline but it seems it is not compared.
- It is not clear if the multi-task baseline is trained in the same way, i.e. with all the 24 losses in the auxiliary space and updating based on a random subset of 24 losses (which introduces some regularization from overfitting). Improvements over the existing multi-task results appear minor.
- When scaling on the number of auxiliary objectives, the authors find no improvement. This is disappointing as one would expect that trivially taking the cartesian product of the space of choices will yield a large number of uninteresting losses and the approach is incapable of pruning them out.

**Summary Of The Paper:**

The paper proposes an approach using meta-learning to learn to weight multiple auxiliary losses for any given target task. They first define a space of auxiliary losses that contains popular self-supervised methods like BERT and XLNet as special cases. Then they propose to learn the weighting over a given space of such losses using the META-TARTAN algorithm from previous work. Approach is evaluated on 5 classification tasks where the dynamic weight learning on auxilary losses improves over a multi-task baseline on 4 out 5 datasets by.

**Summary Of The Review:**

I am not entirely convinced of the utility of this method and it lacks some important baseline comparisons (see weaknesses above).

---

> ### Author Response · Authors · 2022-11-10
> **Response to Reviewer KWWF**
>
> We would like to thank the reviewer for committing time to provide feedback on our paper. We address specific concerns below
>
> > “Differences from the META-TARTAN paper are not clear and are not explicitly mentioned. It should be an important baseline but it seems it is not compared to”
>
> **Please note that Meta-Tartan as presented in Dery et al 2021b is presented as (Z) BERT-Style in Table 2 of our paper**.  We apologise that this was not clear, we have updated the paper to make it more explicit that (Z) is the Dery et al 2021b  baseline. We obtain improvements over Dery et al 2021b ( [Z] Bert-style in Table 2) in 4 out of the 5 tasks we experimented with. Figure 3 also features improvements of AANG over META-TARTAN variants when external data is involved.  Also, please see our general response about our specific contributions beyond Dery et al 2021b. We have also added Appendix B (in blue) to discuss META-TARTAN further
>
> ---
>
> >“How is the multi-task baseline trained? … with all the 24 losses in the auxiliary space and updating based on a random subset of 24 losses (which introduces some regularization from overfitting)”
>
> Our static multitask baseline is trained exactly the same way as with the meta-learning variants of AANG. Specifically, we train the multitask baseline by setting the (meta) learning rates
> of the factors to 0 so they are not changed from their initial, equal weight setting.
> **We cross-validate the number of tasks that are sub-sampled in the same space as for our AANG algorithm (We use the hyperparameters in Table 5 - but we set sopt_lr and aux_lr to 0). Thus the regularising effect of sub-sampling tasks cannot account for the superior performance of meta-learning adaptive weights over static multitask weights (Table 2).**
> We have updated the text of the paper (highlighted in blue) to make this explicit.
>
> ---
>
> >“In particular, I do not find this setup to be specially illuminating, or inspiring new losses just by its construction (as one would not trivially expect every possible choice of loss in this space to be useful)”
>
> We agree with the reviewer that **a-priori, we would not expect all tasks in the constructed search space to be useful to the end-task**. However, **we do expect some tasks to be helpful to the end-task** but since we do not know these a-priori, we are left with the approach of exhaustive enumeration and subsequently adaptively selecting the best tasks via meta-learning.
> Please note that our meta-learning of adaptive task weights is introduced exactly to solve this problem – to give a sense of which tasks are useful to the end-tasks and which are not. Indeed, Figures 4 and 5 show that our algorithm distinguishes the impact of different tasks on the end-task. The performance improvement of meta-learning over static multitasking (Table 2) suggests that our algorithm is learning to upweight useful auxiliary tasks and down-weight harmful ones.
>
> ---
>
> > “The approach seems novel, though it builds and relies heavily on META-TARTAN. The new thing seems to be about the change in dynamic weight updates and it would be good to clarify these”
>
> Please see our general response about specific contributions of our paper that go beyond Meta-Tartan. In summary
> META-TARTAN as an algorithm assumes auxiliary tasks are given a-priori. We provide a way to construct the set of auxiliary objectives.
> META-TARTAN as is, does not scale well with more tasks and also does not take into account the relationship between tasks.
> Dery et al 2021b does not theoretically ground META-TARTAN. We do so for our variant of the algorithm.
>
> ---
>
> > “The overall clarity of the paper can be improved by providing more details about the training algorithm”
>
> Thank you for the feedback. We have modified Section 5 (all modifications highlighted blue text) to better detail the training algorithm at a high level.

---

> > ### Author Response · Authors · 2022-11-10
> > **Continued Response**
> >
> > >“When scaling on the number of auxiliary objectives, the authors find no improvement. This is disappointing as one would expect that trivially taking the cartesian product of the space of choices will yield a large number of uninteresting losses and the approach is incapable of pruning them out”
> >
> > The reviewer is right in observing very little gains (0.44% improvement in ACL-ARC) when we scale up the number of tasks from 24 to 40 – we acknowledged this in section 7.2.
> > **Please note that as we mention in section 7.2, this marginal improvement is as a result of the type of dataset used to construct the extra 16 tasks (specifically, these tasks are constructed using out-of-task data)**. Our theory posits that the impact of extra data depends on the similarity of the newly introduced tasks with the primary task. This similarity is very dependent on the degree of mismatch between the task data and heterogenous domain data. **Thus, at a fixed number of auxiliary tasks, we would need much more data to offset the domain mismatch so as to see marked improvement, a point we raised in Section 7.2**.
> > Also in Section 7.2 and later in Section 8, we acknowledge that our inexact search algorithm is not perfect – it's a good enough proof of concept but more work is needed in the future to improve upon it. Given the scale of our contributions as mentioned in the general response, we deemed further attempts to improve the search algorithm as best left for future exploration.
> >
> > ---
> >
> > > “Code is not provided, though there are more experimental details and hyper-parameters in supplementary to”
> >
> > Apologies for forgetting to mention this, but we will definitely be releasing the code and data necessary to replicate our experiments after the review process.

---

> ### Author Response · Authors · 2022-11-14
> **Revision Deadline Approaching**
>
> Hi Reviewer KWWF,
>
> Since the window for modifying our submission is drawing near (Nov 18th) we would appreciate it if you could let us know if our responses have sufficiently addressed your concerns. If they haven’t, please let us know what is lacking so we can work on making changes before the deadline in a few days.
> Thank you.

---

> ### Author Response · Authors · 2022-11-28
> **Engage with us as rebuttal period enters the last legs**
>
> Hi Reviewer KWWF
>
> It would be great if you could engage with us as the rebuttal process enters its last legs.
> **Unfortunately, we have had zero engagement throughout this nearly month long rebuttal process despite having sent multiple diligent reminders**.
>
> Given that a critical part of the summary of your review is an absence of a baseline -- **which we show already exists in the paper and improve upon in settings with and without external data** -- and have taken steps to make clearer in the updated version, we  would appreciate at least an acknowledgement  of this or  a request for followup clarification.
>
> Thank you.

---

> ### Comment · Reviewer_KWWF · 2022-12-02
> **Post Rebuttal**
>
> I have read the author responses as well as the other reviews. I thank the authors for their thorough response which helped answer the questions that were unclear in the initial draft of the work. It is good that Meta-Tartan was included in the experiments. However, my main concerns with the paper were that there is only marginal improvement over multi-task baseline and when scaling the number of losses (which arguably is what the method should be most useful for) the author's find no improvement. Moreover, I don't think the novelty of the unified taxonomy (which is listed as a contribution) is an important contribution and technical contributions seem fairly limited over Meta-Tartan (which are not reflected strongly in experiments either). As such, I retain my original rating assessment.

---

> > ### Author Response · Authors · 2022-12-03
> > **Response to Reviewer KWWF**
> >
> > Thank you for your response to our rebuttal - we appreciate your opinion!
> >
> > With respect to the significance of the taxonomy and technical contributions over META-TARTAN, we realize that opinions may differ on this, but we respectfully disagree (and the other reviewers rBEM and pGCt seem to have also appreciated the novelty and/or significance of this contribution).
> >
> > With respect to our methods providing “no improvement over the multi-task baseline”, we would like to clarify two things:
> > 1. What is called “the multitask baseline” is actually also a method that we proposed in this paper (since the tasks in this baseline are created from our search-space), although the weights of the tasks are not meta-learned
> > 2. We feel that “no improvement” is not an accurate characterization, in fact AANG provides a gain of 1.1% on average over 5 tasks over this multi-task system. Whether this could be considered marginal or not is subjective — we don’t think it is, but recognize that opinions may differ.

---

### Official Review · Reviewer_oyPz · 2022-10-31

**Confidence:** 3
**Correctness:** 3
**Technical Novelty And Significance:** 3
**Empirical Novelty And Significance:** 2
**Recommendation:** 5

**Clarity, Quality, Novelty And Reproducibility:**

Clarity: The paper is well-written and easy-to-understand.


Quality: The paper is technically sound. Certain weakness in evaluation is pointed out in the "Weakness" section above.

Novelty: The reviewer feels that the automated auxiliary learning problem is novel. The technical that the author adopted for solving the problem is marginally novel because it largely depends on META-TRAIN

Reproducibility: Reproducing such type of AutoML-related work is challenging. Since the author has not released the source code, the reviewer is not sure if all the results are reproducible.

**Strength And Weaknesses:**

Strength:
1. The paper studied a novel problem: trying to automatically pick auxiliary loss functions from a large search-space.
2. Experiments show that AANG is able to outperform baselines (Table 2). Figure 4 also shows that AANG is able to identify the important loss functions during the training procedure, and the weighted + META-Train variant outperforms assigning equal weights to each loss function.

Weakness:
1. The computational complexity of AANG is higher due to the meta-learning phase.
2. The reviewer feels that the author needs to compare with advanced pretraining + distillation algorithms, like "[NeurIPS2020] MINILM: Deep Self-Attention Distillation for Task-Agnostic Compression of Pre-Trained Transformers". In the paper, the author only compared with GPT / XLNet / BERT objectives. However, these objectives may not be suitable for continuous pretraining and the author needs to compare with stronger baselines.
3. The final language model adapted via AANG is applicable to a single specific downstream applications. The author may need to study if the final backbone model adapted via AANG can also generalize to other tasks.

**Summary Of The Paper:**

The paper studied the problem of how to take advantage of auxiliary loss functions in continuously adapting a pretrained language model. The author proposed to automatically pick auxiliary loss functions from a search space. The loss functions are reweighted during the learning process based on how they can help the downstream task. Author conducted experiments in four datasets to demonstrate the effectiveness of the AANG algorithm. Ablation study also shows that the algorithm for updating the weight for each loss function is effective and can pick auxiliary losses that are mostly related to the downstream problem.


**Summary Of The Review:**

Voted for weak rejection due to concern in experimental evaluation, and the computational complexity of the AANG algorithm. The problem of automatically search for auxiliary learning tasks that are helpful for adapting the model is novel.

---

> ### Author Response · Authors · 2022-11-10
> **Response to Reviewer oyPz**
>
> We appreciate the time spent by the reviewer to provide feedback on our paper. We address specific concerns below :
>
> >“The computational complexity of AANG is higher due to the meta-learning phase”
>
> We recognize that using meta-learning comes at an extra computational cost. We acknowledge this fact in Section 8 (Limitations and Conclusion) and presented the Static Multitask-TD variant (Table 2) as a more compute-friendly alternative that still leads to significant improvements over comparable baselines like TAPT and vanilla fine-tuning.
> In the case where the extra computational overhead of meta-learning can be borne, we presented the task sub-sampling approach in Section 5 in order to prevent the meta-learning cost from prohibitively scaling with the size of the search space.
> Please see also our general response for more discussions on the computational cost of AANG
>
>
> ---
>
> > “The final language model adapted via AANG is applicable to a single specific downstream applications. The author may need to study if the final backbone model adapted via AANG can also generalized to other tasks”
>
> Though our work (specifically the automatic generation of a large space of auxiliary objectives for pre-training) can be leveraged in the pre-training setting, within the scope of this paper we chose to focus on the setting where we have a single target task of interest that we know a-priori because of the computational burden of task general pre-training.  **We would like to highlight that our setting is more similar/related to "Fine-tuning a pre-trained model” on a specific end-task**. This is by far the most popular setting relevant to everyday practitioners and thus warrants our attention.
>
> ---
>
> > “The reviewer feels that the author needs to compare with advanced pretraining + distillation algorithms, like [NeurIPS2020] MINILM: Deep Self-Attention Distillation for Task-Agnostic Compression of Pre-Trained Transformers”
>
> We would like to emphasise that **we are mainly concerned with the setting where we continue to train a pre-trained model on a specific end-task (not pre-training from scratch or distillation)**. We decided to focus on the above setting since it is more widespread amongst everyday ML practitioners who use pre-trained models but (like us) cannot bear the compute cost of large scale pre-training / distillation.  Thus,  whilst our methods are complementary to advanced-pretraining and distillation (and specifically [4] as cited by the reviewer), these settings are outside of the current scope of our work. We have added more text to the paper (in blue) to clarify the setting that we operate under.
>
> ---
> > “In the paper, the author only compared with GPT / XLNet / BERT objectives. However, these objectives may not be suitable for continuous pre-training and the author needs to compare with stronger baselines”
>
> Please note that in several NLP works like [1, 2, 3], the listed objectives GPT / XLNet / BERT, and particularly the masked language modelling approach, are shown to be quite effective for continued pre-training.  We thus disagree with the assertion that these objectives are not strong baselines. **We however welcome the suggestions from the reviewer about the specific, stronger baselines for our continued training setting**.
>
> ---
>
> > “Since the author has not released the source code, the reviewer is not sure if all the results are reproducible”
>
> Apologies for forgetting to mention this, but we will definitely be releasing the code and data necessary to replicate our experiments after the review process.
>
> [1] Gururangan, Suchin, et al. "Don't stop pretraining: adapt language models to domains and tasks." arXiv preprint arXiv:2004.10964 (2020)
>
> [2] Dery, Lucio M., et al. "Should we be pre-training? an argument for end-task aware training as an alternative." arXiv preprint arXiv:2109.07437 (2021).
>
> [3] Louvan, Samuel, Silvia Casola, and Bernardo Magnini. "Investigating Continued Pretraining for Zero-Shot Cross-Lingual Spoken Language Understanding." CLiC-it. 2021.
>
> [4]MINILM: Deep Self-Attention Distillation for Task-Agnostic Compression of Pre-Trained Transformers

---

> ### Author Response · Authors · 2022-11-14
> **Revision Deadline Approaching**
>
> Hi Reviewer oyPz
>
> Since the window for modifying our submission is drawing near (Nov 18th) we would appreciate it if you could let us know if our responses have sufficiently addressed your concerns. If they haven’t, please let us know what is lacking so we can work on making changes before the deadline in a few days.
> Thank you.

---

> ### Author Response · Authors · 2022-11-28
> **Engage with us as rebuttal period enters the last legs**
>
> Hi Reviewer oyPz
>
> It would be great if you could engage with us as the rebuttal process enters its last legs.
> **Unfortunately, we have had zero engagement throughout this nearly month long rebuttal process despite having sent multiple diligent reminders**.
>
> In our response to your review we emphasized that  **the setting we are in is most similar to finetuning a pre-trained model rather than pre-training a model from scratch**. We highlighted multiple parts of the paper that mention the fact that we are in the continued training setting and so the reviewer's request for a pre-training based baseline is outside the scope we consider.  We also **highlighted multiple ways in which we have attempted to compensate for the computational overhead of meta-learning**.
>
> We  would appreciate at least an acknowledgement  of this or  a request for followup clarification.
>
> Thank you.

---

> ### Author Response · Authors · 2022-12-06
> **Last week of discussion : Engage with us**
>
> Hi Reviewer oyPz
>
> **We have been comprehensive about responding to your review, and we would hope that since we have only a week left in the discussion phase, you could let us know if you have any further clarifying questions.**.
>
> As noted, **the experimental evaluation you requested is outside of the setting that we explore our work in** (both in terms of the practical setting we address for practitioners and our own compute budget availabilities). We hope that in light of this, you will re-evaluate your assessment of our work.
>
> Thank you

---

### Author Response · Authors · 2022-11-10
**General Response**

We would like to thank all reviewers for their insightful questions and feedback. We appreciate the value that your feedback adds to improving our work further.  **Please note that we have uploaded an updated version of the paper where all modifications are highlighted in blue text**. Below, we address some common themes that came up in your reviews.

## Our Contributions and Connections to META-TARTAN [Dery et al 2021b]

Reviewers oyPz  and KWWF expressed concern about the technical novelty of the work beyond META-TARTAN introduced by Dery et al 2021b. We would like to provide some clarification here, which mostly rehashes what we mention in the intro of our paper and section 5.
Compared to Dery et al 2021b, we make three distinct contributions to the field of auxiliary learning. We list them below
1. **Novel Problem Formulation**: As far as we are aware of, we are the first to formulate the problem of automated auxiliary learning.  Specifically, we presented an approach for automatically constructing a suite of auxiliary objectives based on existing objectives. Please note that Dery et al 2021b perform auxiliary learning with only the DAPT/TAPT variants of the BERT objective. They effectively assume that the search space of objectives (the 2 they explore) is given before-hand. Our approach automatically creates the search space.
2. **Theoretical Novelty**: Again, to the best of our knowledge, we are the first work to provide an exploration of why auxiliary learning improves primary task performance via algorithmic stability. Dery et al 2021b, in introducing META-TARTAN do not attempt to give a theoretical characterization of why the algorithm improves end-task performance.
3. **Algorithm Improvements to META-TARTAN**:  Please note that META-TARAN as presented in Dery et al 2021b was used with only 2 auxiliary tasks. When scaling to more tasks, using META-TARTAN naively becomes computationally prohibitive. Specifically, on a search space of N tasks, META-TARTAN requires O(N) order computation per step.  We improve upon this by introducing the task sub-sampling of (k << N) which reduces the compute overhead to O(k). To account for the impact of sub-sampling as an approximation, we introduced the factorised modelling of task weights which allows sharing of information between auxiliary tasks that might themselves be related – “Because we model each w^k via a factored approach, even if an objective is not sampled its weight is implicitly updated” – (quoted from Section 5)
**We have updated the text of the paper in Appendix B (blue text) to provide further discussion on META-TARTAN and to make our differences more clear**

## Computational Cost of AANG
Reviewers pGCt  and rBEM expressed concern about the computational overhead introduced by AANG.  As we mentioned in Section 8 (Limitations and Conclusion), we acknowledge that using meta-learning introduces extra computational overhead because we have to independently compute meta-gradients for each auxiliary task thus requiring O(k) forward-backward operations for k sampled tasks compared to O(1) for static multitasking. To tackle this shortcoming :
1. At the end of Section 5, we introduced a sampling approach that prevents the meta-learning overhead from scaling with the size of the search space. Given a search space of size N, we subsample $k \ll N$ during each training step. In the most extreme case, one can choose $k=1$. This sampling comes at the cost of our algorithm being an approximate search instead of exact. We provide practitioners with the ability to tune the number of sub-sampled tasks (Table 3) based on how much compute they have available.
2. We presented the Static Multitask-TD baseline as an alternative to doing adaptive task selection via meta-learning. Note that this baseline underperforms the meta-learning version on average but still does significantly better than the strong non-meta-learning baselines we evaluated against. Compute-restricted practitioners can use this as an alternative.

---

### Author Response · Authors · 2022-11-20
**Discussion Stage 2**

Hi Reviewers,

Thanks again for providing feedback on our paper. Since the second phase of discussion has started, please let us know if our updated paper and responses have clarified your questions / concerns.

**We hope that our responses to your concerns, warrant a re-evaluation of your initial reviews in light of the added information or at least inspire engagement with us for further clarification.**

Thank you.

---

### Decision · Program_Chairs · 2023-01-20

**Decision:**

Accept: notable-top-25%

**Justification For Why Not Higher Score:**

The paper makes general claims, but focuses only on language. More experiments in other domains would have shed a light on that.

**Justification For Why Not Lower Score:**

The paper has an interesting idea and findings that will potentially be useful to other researchers. Two of the three reviewers felt that the paper is quite strong.

**Metareview: Summary, Strengths And Weaknesses:**

This paper concerns ways to take advantage of auxiliary losses when training/adapting a pretrained model. The paper categorizes losses along several dimensions and proposes a method to automatically pick losses based on downstream task performance.

The paper offers a way to organize the plethora of auxiliary losses out there and an interesting idea of how to train in auto ML settings. The paper shows empirical improvements on four out of five datasets.

Some of the exposition can be improved as parts are laborious to read through. The ideas are only tested on language tasks -- it will be a lot more convincing if the paper demonstrated that the same ideas work in other domains.

**Note From Pc:**

if the above contains the word "oral" or "spotlight" please see: "oral" presentation means -> notable-top-5% and "spotlight" means -> notable-top-25%. As stated in our emails, we are disassociating presentation type from AC recommendations